# Comparative De Novo and Pan-Genome Analysis of MDR Nosocomial Bacteria Isolated from Hospitals in Jeddah, Saudi Arabia

**DOI:** 10.3390/microorganisms11102432

**Published:** 2023-09-28

**Authors:** Molook Alghamdi, Effat Al-Judaibi, Mohammed Al-Rashede, Awatif Al-Judaibi

**Affiliations:** 1Department of Biological Sciences, Microbiology Section, Faculty of Science, Jeddah University, Jeddah 21959, Saudi Arabia; molook.ma@gmail.com (M.A.); eaaljedeibi@uj.edu.sa (E.A.-J.); 2Maternity and Children Hospital Almosadya, Jeddah 21425, Saudi Arabia; alrasheddi@hotmail.com

**Keywords:** MDR, antibiotics, resistance, *E. coli*, *K. pneumoniae*, *S. aureus*

## Abstract

Multidrug-resistant (MDR) bacteria are one of the most serious threats to public health, and one of the most important types of MDR bacteria are those that are acquired in a hospital, known as nosocomial. This study aimed to isolate and identify MDR bacteria from selected hospitals in Jeddah and analyze their antibiotic-resistant genes. Bacteria were collected from different sources and wards of hospitals in Jeddah City. Phoenix BD was used to identify the strains and perform susceptibility testing. Identification of selected isolates showing MDR to more than three classes on antibiotics was based on 16S rRNA gene and whole genome sequencing. Genes conferring resistance were characterized using de novo and pan-genome analyses. In total, we isolated 108 bacterial strains, of which 75 (69.44%) were found to be MDR. Taxonomic identification revealed that 24 (32%) isolates were identified as *Escherichia coli*, 19 (25.3%) corresponded to *Klebsiella pneumoniae*, and 17 (22.67%) were methicillin-resistant *Staphylococcus aureus* (MRSA). Among the Gram-negative bacteria, *K. pneumoniae* isolates showed the highest resistance levels to most antibiotics. Of the Gram-positive bacteria, *S. aureus* (MRSA) strains were noticed to exhibit the uppermost degree of resistance to the tested antibiotics, which is higher than that observed for *K. pneumoniae* isolates. Taken together, our results illustrated that MDR Gram-negative bacteria are the most common cause of nosocomial infections, while MDR Gram-positive bacteria are characterized by a wider antibiotic resistance spectrum. Whole genome sequencing found the appearance of antibiotic resistance genes, including *SHV*, *OXA*, *CTX-M*, *TEM*-1, *NDM*-1, *VIM*-1, *ere*(A), *erm*A, *erm*B, *erm*C, *msr*A, *qac*A, *qac*B, and *qac*C.

## 1. Introduction

Antibiotic resistance develops from the abuse and overuse of antibiotics. Specifically, a selection of bacteria resistant to most forms of antibiotics emerges from the widespread and unjustified use of broad-spectrum antibiotics [1]. In a clinical environment, such as in hospitals, the overuse of antibiotics, even under controlled administration, breeds antibiotic resistance. More alarming is the presence of multidrug-resistant (MDR) bacteria, which refers to bacteria that exhibit resistance to at least three different classes of antimicrobials [2]. MDR bacteria represent a major global health problem. Infections of MDR bacteria are associated with unfavorable prognoses, even in clinically managed cases, resulting in financial stress to both the patients and the healthcare system [2]. There has been an increase in antibiotic resistance despite the increase in commercially available antibiotics over the past few decades. In response, the World Health Organization (WHO) [3] has issued an urgent demand for new treatments against the world’s leading antibiotic-resistant bacteria [4]. Specifically, research into innovative drugs (i.e., drugs with an active ingredient that does not demonstrate cross-resistance to existing antibiotics) is warranted [2,4]. However, developing an active substance requires understanding the mechanisms of antibiotic resistance. The antibiotic resistance of MDR bacteria results from antibiotic inactivation through one or a combination of molecular mechanisms. One of the most documented mechanisms of resistance to antibiotics is the production of enzymes that inactivate their biological activity. For example, β-lactamase inactivates the β-lactam ring, the primary active component of several antibiotics that are classified as β-lactams, such as penicillins, cephalosporins, carbapenems, and monobactams [5]. These β-lactam antibiotics target penicillin-binding proteins (PBPs) by acylating the transpeptidase, resulting in inhibition of the final processes of peptidoglycan synthesis [6]. *K. pneumoniae* carbapenemase also hydrolyzes and inactivates a specific class of β-lactams, namely, the carbapenems. Antibiotic-resistant *S. aureus* and *E. coli* also produce carbapenemase. In addition to β-lactamase, enzymes that inactivate antibiotic activity include acetyltransferase, phosphotransferase, and adenyltransferase [2]. Altering the antibiotic target without interrupting cellular processes represents another mechanism of antibiotic resistance. For instance, methicillin-resistant *S. aureus* (MRSA) carries *mecA*, which encodes the novel penicillin-binding protein PBP2a, which differs from the PBP targeted by older β-lactam antibiotics, rendering them ineffective against MRSA [7]. Notably, staphylococcal cassette chromosome (SCC*mec*) of classes I, II, III, or IV was acquired by horizontal gene transfer across hospital-acquired MRSA (HA-MRSA) strains of different lineages [7]. The horizontal transfer of genes that confer antibiotic resistance to pathogens represents an emerging cause of concern for public health [8]. This threat already exists for HA-MRSA, one of the most common hospital-acquired infections caused by MDR bacteria [9].

MRSA produces aggressive biofilms, reducing the permeability of the bacteria [10]. Reducing cellular permeability is a mechanism of antibiotic resistance characterized by alterations in the structure of the cell surface casing [11]. In the case of MRSA and *P. aeruginosa*, aggressive biofilm production increases their resistance to the host immune system as well as antibiotics [9]. According to the Ministry of Health (MOH), a work plan is in place because of the threat of antimicrobial resistance to public health, necessitating various activities, and coordination between government agencies, the private sector, and the public [11]. Compared to the rates in the 1990s in Saudi Arabia, during the subsequent decade, a considerable increase in the prevalence of carbapenem-resistant Gram-negative bacteria was observed, specifically of strains that produce extended-spectrum beta-lactamase (ESBL). At the same time, some institutes have reported ESBL rates of 29% for *E. coli* and 65% for *K. pneumoniae* of all isolated MDR bacteria. These rising rates have been linked to several outbreaks and fatalities ranging from 11% to 40%. According to Saudi Arabia’s national surveillance on Gram-positive cocci, 32% of *S. aureus* strains are methicillin-resistant (MRSA), and 33% and 26% of *S. pneumoniae* strains are resistant to penicillin G and erythromycin, respectively [12].

Jeddah is the second largest city in Saudi Arabia, with a population of 3.2 million people. Jeddah is commercially important and serves as the entry point to the holy cities of Makkah and Medina [13]. According to studies conducted in three private hospitals and hospitals of MOH in Jeddah, among the Gram-negative MDR bacteria, species of the Enterobacteriaceae family were the most frequently isolated, with *E. coli* being the most common organism, representing 40.0% [14,15]. Regarding Gram-positive MDR bacteria, a study conducted on the incidence of Gram-positive MDR bacteria at King Abdulaziz University Hospital, Jeddah, found that the most prevalent isolates (20%) were *S. aureus* (MRSA) [16].

This study aims to identify MDR isolates based on the definition of MDR bacteria as those that are resistant to at least three or more antibiotic classes.

## 2. Materials and Methods

### 2.1. Ethical Approvement

This study did not require ethical approval, as no human subjects were involved.

### 2.2. Clinical Specimens

A total of 108 isolates were isolated from different types of medical specimens (stool; sputum; blood; urine; high vaginal swabs; and swabs from wounds, ears, eyes, noses, etc.) and from several hospitals wards, e.g., the neonatal intensive care unit, intensive care unit (ICU), emergency room, and outpatient department (OPD). The isolates were collected from three hospitals (MOH) between December 2018 and November 2019. All of the isolates from each hospital laboratory were grown on blood agar and then transferred to the microbiology laboratory at Jeddah University for further studies.

### 2.3. Isolation and Identification of MDR Bacteria

The combination panels of the BD Phoenix M50 identification system (NMIC/ID-26; catalog no. 448026) were used for phenotype identification and antibiotic susceptibility testing of the bacterial isolates, in accordance with the manufacturer’s recommendations. This system uses a redox indicator to detect the growth of organisms in the presence of an antimicrobial agent [17]. The bacteria were identified at the Maternity and Children’s Hospital in Jeddah.

### 2.4. Amplification of 16S rRNA Gene, Sequencing, and Phylogenetic Analysis

Seven MDR bacterial DNA samples were extracted using the Qiagen DNA extraction kit (Qiagen, Hilden, Germany), according to the manufacturer’s instructions [18]. The extracted DNA was stored at −20 °C.

The 16S rRNA gene was amplified by PCR using the following couples of forward and reverse primers: 518F 5′-CCAGCAGCCGCGGTAATACG-3′ and 800R 5′-TACCAGGGTATCTAATCC-3′; 27F 5′-AGAGTTTGATCMTGGCTCAG-3′ and 1492R 5′-GGTTACCTTGTTACGACTT-3′.

The primers used for the amplification of the 16S rRNA gene were designed based on the conserved regions in the 16S rDNA sequences [19]. DNA sequencing was conducted by MiSeq (Illumina) using the v3 chemistry (600 cycles); sequencing of amplification products was carried out by the Infection Disease Unit, King Fahad Medical Center for Research, King Abdulaziz University 8/2/2021, and the sequenced data were analyzed using BLAST-National Center for Biotechnology Information (NCBI) (available online: http://blast.ncbi.nlm.nih.gov/Blast.cgi (accessed on 29 August 2021)). The phylogenetic tree diagram of the seven isolated MDR bacteria samples based on 16S rDNA sequencing was constructed using the maximum likelihood (ML) algorithm in MEGA7 software (version 10.2, megasoftware.net (accessed on 2 September 2021)). Node stability was assessed by bootstrapping 1000 replicates. The accession number of the 16S rDNA gene of each strain was obtained from the National Center for Biotechnology Information (NCBI) (http://blast.ncbi.nlm.nih.gov/Blast.cgi (accessed on 29 August 2021)).

### 2.5. Whole Genome Sequencing and Analysis Workflow

Genomic DNA was extracted from the seven isolated bacteria samples according to the protocol provided by the Qiagen genome DNA extraction kit. After completion of quality control, the selected isolates were used for library TruSeq Nano DNA (350), the run type Throughput Run, in 36 Scale. For cluster generation, the library was loaded onto a flow cell, where fragments were captured on a lawn of surface-bound oligos complementary to the library adapter sequences and proceeded using an average 350 bp insert size with “covaries”. Each fragment was amplified into distinct clonal clusters through bridge amplification. When cluster generation was complete, the templates were ready for sequencing. As all four reversible, terminator-bound dNTPs are present during each sequencing cycle, natural competition minimizes incorporation bias and greatly reduces the raw error rates compared with other technologies. This results in highly accurate base-by-base sequences where sequence-context-specific errors are virtually eliminated, even within repetitive sequence regions and homopolymers. After sequencing, the identified bases and the predicted quality of each base were converted into raw data for analysis. In this step, BCL/cBCL files were converted into FASTQ (http://www.bioinformatics.babraham.ac.uk/projects/fastqc (accessed on 15 May 2022)) files using the Illumina package bcl2fastq [20].

### 2.6. De Novo Assembly of MDR Bacterial Sequences

For the de novo assembly, the overlap–layout–consensus (OLC) algorithm was adopted, which identifies overlaps between reads, with the layout of the reads and building of consensus sequences of the read sequences generated by the Illumina sequencing platform (Novaseq-6000) with the De Bruijn graph (DBG) algorithm. The total read bases were *Pseudomonase*_1 “3.526 GB”, *E. coli*_7 “2.774 GB”, *K. pneumoniae*_3 “3.047 GB”, *K. pneumoniae*_6 “2.918 GB”, *S. aureus*_9 “3.508 GB”, *S. aureus*_10 “2.313 GB”, and *S. aureus*_12 “2.254 GB”. After de novo assembly was achieved, the contigs (*Pseudomonas*_1 “100”, *E. coli*_7 “47”, *K. pneumoniae*_3 “45”, *K. pneumoniae*_6 “50”, *S. aureus*_9 “48”, *S. aureus*_10 “20”, and *S. aureus*_12 “28”) were generated with N50, and the generated reads of the size range of bp were as follows: *Pseudomonase*_1, 669, *E_coli*_7, 632, *K_pneumoniae*_3, 652, *K_pneumoniae*_6, 641, *S_aureus*_9, 653, *S_aureus*_10, 664, and *S_aureus*_12, 652. These were assigned a taxonomic classification based on alignment using the nucleotide Basic Local Alignment Search Tool (BLASTN) against the nucleotide collection database (NCBI database). Following quality trimming, trimmed paired-end reads were de novo assembled into contigs. Contigs with a total read count of >1000 reads were chosen for alignment with software that used the DBG algorithm, including SPAdes version 3.15.4 (http://cab.spbu.ru/software/spades/ (accessed on 15 May 2022)), Platanus, and SOAPdenovo, for bacterial identification. The DBG algorithm divides each read into pieces with a length K, referred to as K-mers, and progressively extends them by 1 base to select the most expected sequence [21,22].

### 2.7. Pan-Genome Analysis

#### 2.7.1. Calculation of Mash Distances and ANI

To identify the taxonomic boundary of the genomes, average nucleotide identity (ANI) was calculated using the Mash tool [21]. The ANI calculator (https://www.ezbiocloud.net/tools/ani (accessed on 15 May 2022)) was used to compare two prokaryotic genomes based on ANI values calculated for both genomes, followed by the classification and identification of different bacteria. The pan-genome was used as a reference genome in all cases.

For quality control assessment and to ensure error-free sequences, the genomic sequences were subjected to Panaroo [23]. This helped to identify the Mash distances between the assemblies by using Mash. Mash calculates the dissimilarities among the genomes as well as genome-scale similarity, referred to as ANI [21]. To interpret the results, Pheatmap in R was used [24]. Pan-genome analysis was performed by Macrogen, Seoul, Republic of Korea (https://www.macrogenusa.com/ (accessed on 22 July 2022)).

#### 2.7.2. Profiling of Virulence and Antibiotic Resistance Genes

ABRicate v0.8 (https://github.com/tseemann/abricate (accessed on 22 July 2022)) was used to identify virulence and antibiotic resistance genes and characterize their presence and absence.

## 3. Statistical Analysis

Statistical analyses were performed using the Statistical Package for the Social Sciences, version 20 (IBM, Armonk, NY, USA). The results are expressed as the mean ± standard deviation. Differences between samples and homogeneity between groups were determined using ANOVA. The differences were considered significant at *p* ≤ 0.05.

The total number of base reads, GC (%), Q20 (%), and Q30 (%) were calculated for each isolate. The phylogenetic indices that serve as a quantitative index for the number of species are found in a dataset (a community), as are the phylogenetic relationships (co-distribution, species affinity, and species richness) between distributed individuals (Table 1). After sequencing, the FastQC program was used for quality control. According to quality control results, the number of data read qualities, GC distributions, k-mer distributions, and possible adapter contamination were determined and evaluated for each sample [25]. After quality control, reads with poor quality (Phred score < Q20, 30 bp window range) were removed from the data. Furthermore, low-quality base reads at terminal regions and chimeric sequences with possible adapter contaminants were removed using the Trimmomatic tool and Genomes OnLine Database (GOLD) [26]. Taxonomic profiling was performed using Kraken2 [27], and the Silva (2020) database was used as a reference dataset [28]. The OTU groups in each sample were determined after alignment. R scripts were used for data reporting, statistical analyses, and data visualization (https://www.R-project.org/ (accessed on 22 July 2022)).

## 4. Results

A total of 108 isolates were analyzed in this study, of which 75 were MDR bacteria. In terms of patient age, the number of bacterial isolates was 41 (37.97%) from 0–10-year-olds, 10 (9.26%) from 11–20-year-olds, 20 (18.52%) from 21–30-year-olds, 19 (17.59%) from 31–40-year-olds, and 18 (16.67%) from >41-year-olds. Gender distribution analysis showed that 32 (29.63%) strains were isolated from male patients and 76 (70.37%) from female patients.

### 4.1. Identification of Isolated Bacteria

Phenotype analysis of the 108 bacterial isolates identified 33 (30.56%) as *E. coli*, 27 (25%) as *K. pneumoniae*, 2 (1.85%) as *K. aerogenes*, 9 (8.33%) as *P. aeruginosa*, 3 (2.78%) as *E. cloacae*, 2 (1.85%) as *Salmonella enterica*, 20 (18.52%) as *S. aureus*, 1 (0.93%) as *S. epidermidis*, 2 (1.85%) as *Streptococcus pneumoniae*, 8 (7.41%) as *S. agalactiae* (B), and 1 (0.93%) as *Acinetobacter baumannii*. The AST results identified one isolate of *P. aeruginosa*, three isolates of *E. coli*, four of *K. Pneumoniae*, and three of *S. aureus* as being resistant to more than three classes of antibiotics. 

Of the 108 isolates, 75 (69.44%) were identified as MDR bacteria. These isolates were predominately represented by the following strains: *S. aureus* (MRSA), 17 (22.67%); MDR *S. epidermidis*, 1 (1.33%); MDR *S. pneumoniae*, 2 (2.67%); ESBL-producing Gram-negative Enterobacteriaceae, such as ESBL *E. coli*, 24 (32%); ESBL *K. pneumoniae*, 19 (25.33%); *K. aerogenes* 1(1.33%); *E. cloacae*, 2 (2.67%); and MDR *P. aeruginosa*, 9 (12%). Following this initial screening step, we selected two *E. coli* isolates in addition to two *K. pneumoniae* and three *S. aureus* isolates to perform the remaining investigations.

### 4.2. Identification Based on 16S rRNA Gene and Whole Genome Sequencing

The genus-level identification based on the top five hits of the contigs ordered by length and use of the NCBI NT database of the seven selected isolates confirmed the preliminary molecular identification results based on 16S rRNA gene sequencing, except for isolates of *P. aeruginosa*_1, whose identification differed from that indicated by the biochemical and preliminary molecular tests. The isolated MDR bacteria were identified as members of the genera *Klebsiella* (53.10%), *Escherichia* (42.48%), *Citrobacter* (1.77%), *Salmonella* (1.77%), and *Enterobacter* (0.88%) (Figure 1a). Furthermore, the identity of *E. coli*_7 and *K. pneumoniae*_3 was confirmed on the basis of 16S rDNA sequence identities of 87.23% and 75.56%, respectively. The isolates *K. pneumoniae*_6, *S. aureus*_9, *S. aureu*s_10, and *S. aureu*s_12 showed 100% identity at the genus level (Figure 1b–g). 

### 4.3. Phylogenetic Tree of the Seven Selected MDR Bacteria

The phylogenetic tree shows the multiple alignments of MDR genome contigs sequences of the seven isolated species (Figure 2). The results in Figure 3 each isolate were used to assess the completeness of genome assembly using Benchmarking Universal Single-Copy Orthologs (BUSCO) analysis and ANI. Nucleotide-level genomic similarity between the species from the same taxon available in the NCBI database was used as the measurement unit to determine the accession number for each isolate. For *Pseudomonas*_1, the rate of complete single-copy BUSCOs (S) was 6.45%, and 93.55% had complete and duplicated BUSCOs (D) (Figure 3a1), whereas the top five genomes identified by ANI analysis were GCA_023366235.1_*E. coli*_PNUSAE100989, with an ANI of 98.87% and 39.89% coverage by the assembly sequence alignments against the compared genome; GCA_023366335.1_*E. coli*_1193, with an ANI of 98.82% and 41.29% coverage; GCA_023367035.1_*E. coli*_131, with an ANI of 98.55% and 42.62% coverage; GCA_023367635.1_*E. coli*_131, with an ANI of 98.55% and 42.39% coverage; and GCA_023368475.1_*E. coli*_131, with an ANI of 98.54% and 42.41% (Figure 3b1,c1). *E. coli*_7 had 100% complete single-copy BUSCOs (S) (Figure 3a2), whereas the top five genomes identified by ANI analysis were GCA_023368155.1_*E.coli*_4532, with an ANI of 99.26% and 89.32% coverage; GCA_023367075.1_*E. coli*_4532, with an ANI of 99.26% and 89.29% coverage; GCA_023367595.1_*E. coli*_, with an ANI of 99.19% and 92.15% coverage; GCA_023369735.1_*E. coli*_DC5_C10, with an ANI of 99.13% and 88.78% coverage; and GCA_023367495.1_*E. coli*_191, with an ANI of 99.12% and 88.78% coverage (Figure 3b2,c2). *K. pneumoniae*_3 had 98.39% complete single-copy BUSCOs (S) and 1.61% missing BUSCOs (M) (Figure 3a3), whereas the top five genomes identified by ANI analysis were GCA_023368715.1_*K. pneumoniae*_11, with an ANI of 99.30% and 86.83% coverage; GCA_023368275.1_*K. pneumoniae*_11, with an ANI of 99.29% and 88.69% coverage; GCA_023368575.1_*K. pneumoniae*_1161, with an ANI of 99.29% and 87.91% coverage; GCA_023367775.1_*K. pneumoniae*_1, with an ANI of 99.29% and 88.16% coverage; and GCA_023367655.1_*K. pneumoniae*_133, with an ANI of 99.29% and 87.19% coverage (Figure 3b3,c3). In addition, *K. pneumoniae*_6 had 98.39% complete single-copy BUSCOs (S) and 1.61% missing BUSCOs (M) (Figure 3a4), whereas the top five genomes identified by ANI analysis were GCA_023369875.1_*K. pneumoniae*_13, with an ANI of 99.94% and 94.96% coverage; GCA_023369885.1_*K. pneumoniae*_14, with an ANI of 99.94% and 94.44% coverage; GCA_023370005.1_*K. pneumoniae*_6, with an ANI of 99.93% and 95.29% coverage; GCA_023370035.1_*K. pneumoniae*_2, with an ANI of 99.93% and 98.04% coverage; and GCA_023369995.1_*K. pneumoniae*_3, with an ANI of 99.93% and 98.0% coverage (Figure 3b4,c4).

Gram-positive *S. aureus*_9 had 100% complete single-copy BUSCOs (S) (Figure 3a5), whereas the top five genomes identified by ANI analysis were GCA_023361315.1_*S. aureus*_, with an ANI of 99.13% and 94.31% coverage by assembly sequence alignments against the compared genome; GCA_023361575.1_*S. aureus*_, with an ANI of 99.11% and 94.17% coverage; GCA_023363605.1_*S. aureus*_, with an ANI of 99.11% and 93.82% coverage; GCA_023361695.1_*S. aureus*_, with an ANI of 99.10% and 93.77% coverage; and GCA_023361615.1_*S. aureus*_, with an ANI of 99.10% and 93.71% coverage (Figure 3b5,c5). *S. aureus*_10 had 98.39% complete single-copy BUSCOs (S) (Figure 3a6), whereas the top five genomes identified by ANI analysis were GCA_023362615.1_*S. aureus*_, with an ANI of 99.93% and 98.42% coverage; GCA_023362315.1_*S.aureus*_, with an ANI of 99.13% and 93.37% coverage; GCA_023362475.1_*S. aureus*_, with an ANI of 99.10% and 93.98% coverage; GCA_023361575.1_*S. aureus*_, with an ANI of 99.09% and 92.55% coverage; and GCA_023361495.1_*S. aureus*_, with an ANI of 99.08% and 93.22% coverage (Figure 3b6,c6). *S. aureus*_12 had 100.00% complete single-copy BUSCOs (S) (Figure 3a7), whereas the top five genomes identified by ANI analysis were GCA_023360955.1_*S.aureus*_, with an ANI of 99.87% and 94.39% coverage; GCA_023362575.1_*S.aureus*_, with an ANI of 99.87% and 93.26% coverage; GCA_023361655.1_*S. aureus*_, with an ANI of 99.86% and 94.67% coverage; GCA_023362455.1_*S. aureus*_, with an ANI of 99.86% and 96.15% coverage; and GCA_023362395.1_*S. aureus*_, with an ANI of 99.83% and 96.75% coverage (Figure 3b7,c7).

### 4.4. Detection of MDR Genes

The MDR genes were detected in *Pesudomonas*_1, *E. coli*_7, *K. pneumonia*_3, *K. pneumonia*_6, *S. aureus*_9, *S. aureus*_10, and *S. aureus*_12. The results are illustrated in Figure 4. *SHV*, *OXA*, *CTX-M*, *TEM*-1, *NDM*-1, *VIM*-1, and *ere*(A) genes were detected in *Pesudomonas*_1, *E. coli*_7, *K. pneumonia*_3, and *K. pneumonia*_6, whereas *erm*A, *erm*B, *erm*C, *msr*A, *qac*A, *qac*B, and *qac*C genes were detected in *S. aureus*_9, *S. aureus*_10, and *S. aureus*_12. All genomic information is available in the Appendix A (Appendix A). Venn diagram showing the susceptibility of multidrug-resistant bacteria isolated from selected hospitals in Jeddah to 108 universal antibiotics, the result shows that *Pseudomonas*_1 and *E. coli*_7 are resistant to 34 antibiotics, while *S. aureus*_9, *S. aureus*_10, and *S. aureus*_12, are resistant to 15 antibiotics (Figure 5, data on antibiotics are available in Appendix A).

## 5. Discussion

MDR is defined as resistance to antimicrobial drugs from at least one to three or more categories. Hospitals are an ideal environment for bacterial colonization and a potential source of bacterial infection, thereby increasing the possibility of infection, particularly by MDR nosocomial bacteria. Thus, conducting studies on MDR nosocomial bacteria is necessary to control and prevent infection in hospitals.

With the emergence of MDR bacteria as a global public health concern, a novel approach for the management of MDR bacteria is warranted. Identifying these MDR bacteria and comparing them with the global isolates determines the extent of the spread of different species and provides a statement of new species, thereby allowing the identification and inventory of species in each country and of the common species among different countries. By comparing the isolated species with species published in other research, a similarity is observed between the species in our study, including *E. coli*, *K. pneumoniae*, and *S. aureus*, and the species isolated from different countries in several publications [29,30,31,32].

Antibiotic resistance may be a result of an energy-driven efflux system within the cell that removes foreign antibiotic agents from the cell. For instance, the resistance of pathogens, including *K. pneumoniae* and *E. coli*, to tetracyclines is caused by the acquisition of the *tet*(*A*) and *tet*(*B*) genes by Gram-negative bacteria and *tet*(*K*) and *tet*(*L*) genes by Gram-positive bacteria. In this study, Gram-negative bacteria were resistant to multiple antibiotics, in addition to tetracyclines, due to the presence of *SHV*, *OXA*, *CTX-M*, *TEM*-1, *NDM*-1, and *VIM-1* genes. These results are consistent with those reported in other studies [33,34,35,36]. Gram-positive bacteria carried resistance owing to the presence of *erm*A, *erm*B, *erm*C, *msr*A, *qac*A, *qac*B, and *qac*C genes. Our results are in accordance with other reports detecting the presence of the genes *erm*B [37]; *erm*A, *msr*A, and *qac*A [8,38,39]; and *erm*A, *erm*B, and *erm*C [40] in *S. aureus*. These genes encode efflux pumps [2]. However, antibiotic resistance can also emerge from the activation of an alternative metabolic pathway in the presence of an antibiotic that inhibits a cellular process. For instance, sulfamidics that inhibit folic acid synthesis are ineffective against bacteria that have the ability to synthesize folic acid through alternative metabolic pathways [41]. 

The increase in the spread of infections by isolated MDR bacteria in the winter season may be a result of frequent visits to hospitals because of diseases associated with winter; in this respect, the results of our study agree with those of other reports [42,43,44,45]. Interestingly, the bacteria occurring most often in winter are respiratory pathogens, such as *S. aureus* MRSA. This could be due to resistance, stability in dehydrating conditions, and presence in the ventilation streams of air conditioners [46,47,48].

To examine contamination, inanimate objects and patient-care equipment were sampled for MDR bacteria because pathogenic bacteria tend to accumulate in service locations, such as restrooms and canteens. *S. aureus* has the characteristic ability to survive in dry conditions and both high and low temperatures. The difference in the prevalence of MDR bacteria in different samples was found to be caused by variable exposure to antibiotics [49].

Antibiotics are becoming ineffective because of MDR characteristics. In this study, we report the presence of MDR in all groups of bacteria. The genera that were identified by de novo analysis were *Klebsiella*, *Escherichia*, and *Staphylococcus*. The species that were identified were *Pseudomonas*_1, *E. coli*_7, *K. pneumoniae*_10, *S. aureus*_19, and *S. aureus*_12. There were no significant differences in antibiotic resistance in the isolated strains of *E. coli, K. pneumoniae*, *K. aerogenes, E. cloacae*, and *P. aeruginosa* for all tested antibiotics, except gentamicin in *E. cloacae*. Several studies have produced results in accordance with our findings [50,51,52,53]. The increase in the resistance of *E. coli*, *Pseudomonas*, and *S. aureus* to antibiotics may have been caused by the indiscriminate use of antibiotics without prescriptions. Many studies have reported the resistance of *E. coli*, *Pseudomonas*, and *S. aureus* to one or more antibiotics and have identified some of the genes responsible for this resistance [54,55,56,57,58].

Gram-positive bacteria are more sensitive than Gram-negative bacteria, which are more stable to change [59,60,61], with *K. pneumoniae* demonstrating stability in biochemical reactions [62]. A higher rate of resistance to most commonly used antibiotics has been observed [63]. *E. coli* and *K. pneumoniae* were the most frequently isolated among Gram-negative MDR bacteria, with high resistance rates. These findings are in agreement with those of other studies [64,65,66]. The results of *E. coli*, *Pseudomonas*_1, and *K. aerogenes* in our study are in accordance with the results of other investigations [9,67,68]. Regarding Gram-positive isolates, our findings illustrated that *S. aureus* MRSA and *S. epidermidis* are resistant to most antibiotics, while *S. pneumoniae* is less resistant to antibiotics as reported by previous other studies [69,70,71].

The result in this study indicates that MDR bacteria represent a healthcare problem in hospitals for patients of all ages and awards. This result may be due to the adaptation of pathogenic bacteria, which leads to the development of mechanisms by which they can survive and acquire antibiotic resistance [72,73,74]. The result of this study is consistent with that of previous studies [12,74,75].

Gram-negative bacteria were more resistant to antibiotics than Gram-positive bacteria because of the differences in the mechanism of gene transfer between Gram-negative and Gram-positive bacteria. In addition, Gram-negative bacteria have an active uptake mechanism under iron-depleted conditions, which improves stability to different types of β-lactamases [76,77]. Bacterial resistance to antibiotics is also due to bacterial mutation, and the appearance of antibiotic-resistant genes in Gram-negative and Gram-positive bacteria can occur through the production of new generations, with a timescale of just a few hours, or new mutations in a parent cell can become a widespread mutation, resulting in the microevolution of resistant colonies [78,79]. Our finding is consistent with that of previous studies [80,81,82,83,84].

## 6. Conclusions

There is an urgent need to identify bacteria that are resistant to more than one antibiotic, to discover antibiotics that inhibit their growth. This study provides valuable insights into the presence of MDR bacteria in hospitals and antibiotic-resistance genes in Gram-positive and Gram-negative bacteria. The results presented here identified strains of *S. aureus*, *E. coli*, *K. pneumoniae*, and *P. aeruginosa* as MDR bacteria isolated from hospitals in Jeddah, Saudi Arabia, and Gram-negative isolates were resistant to antibiotics, in contrast with Gram-positive isolates. Detecting some genes that cause bacterial resistance promotes the development of a mechanism to inhibit the infection caused by these resistant bacteria. Our findings are largely consistent with those of previous studies, but further investigation at other hospitals is needed to assess their generalizability.

## Figures and Tables

**Figure 1 microorganisms-11-02432-f001:**
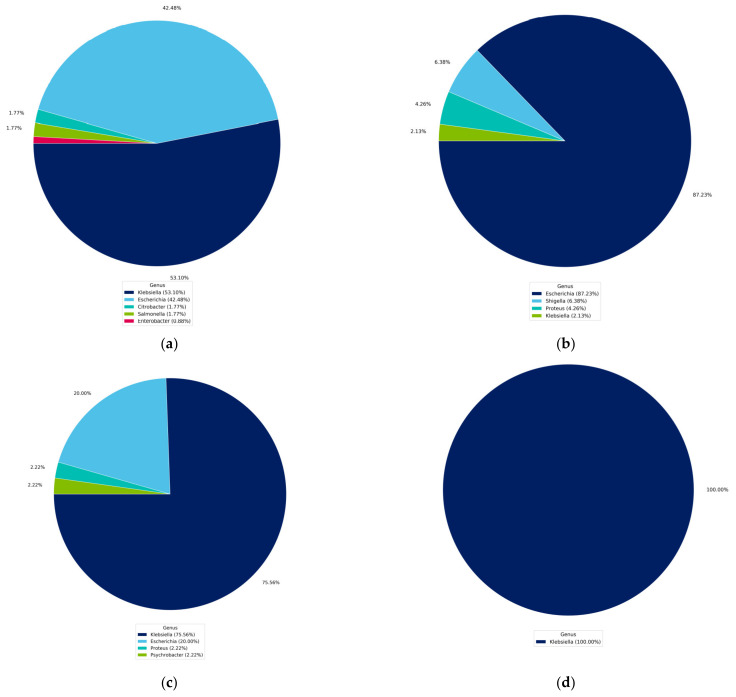
Genus-level summary of de novo assembly and analysis of the RNA-seq data obtained by BLASTN for the isolated MDR bacteria: (**a**) *P. aeruginosa*_1; (**b**) *E. coli*_7; (**c**) *K. pneumoniae*_3; (**d**) *K. pneumoniae*_6; (**e**) *S. aureus*_9; (**f**) *S. aureus*_10; and (**g**) *S. aureus*_12.

**Figure 2 microorganisms-11-02432-f002:**
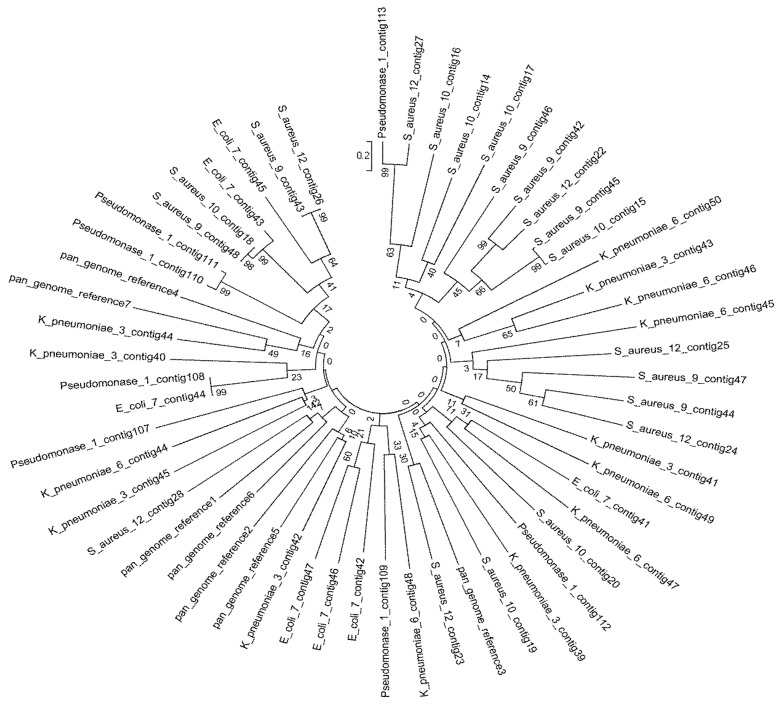
Phylogenetic tree of the MDR bacterial isolates based on contigs sequencing.

**Figure 3 microorganisms-11-02432-f003:**
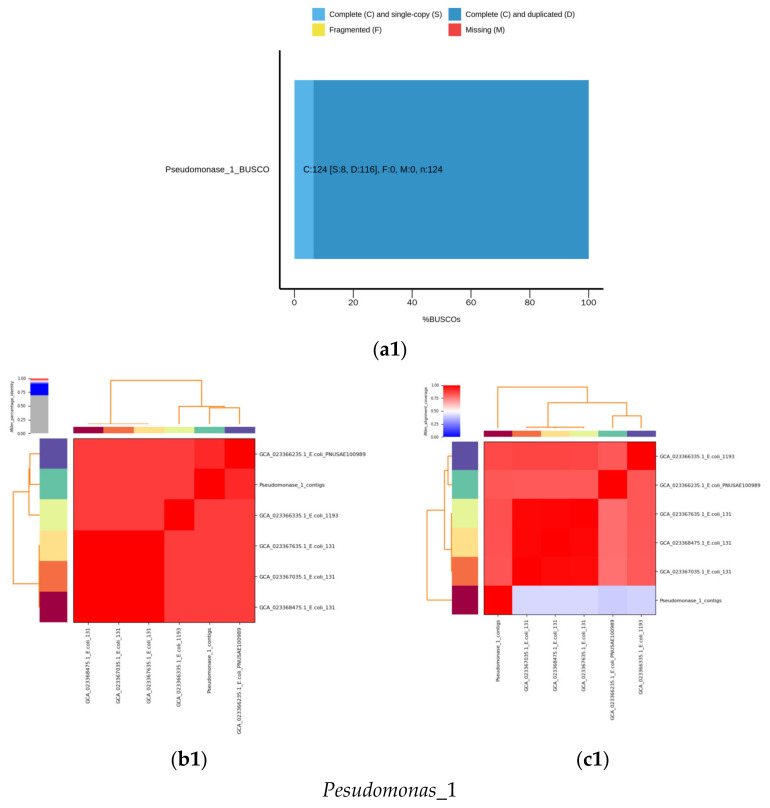
De novo assembly of *Pseudomonas*_1, *E. coli*_7, *K. pneumoniae*_3, *K. pneumoniae*_6, *S. aerueus*_9, *S. aerueus*_10, and *S. aureus*_12 isolated from selected hospitals in Jeddah. (**a**) Plot of BUSCO assessment results. (**b**) Identity heatmaps (%) identified by ANI analysis. (**c**) Heatmaps of alignment coverage from ANI analysis.

**Figure 4 microorganisms-11-02432-f004:**
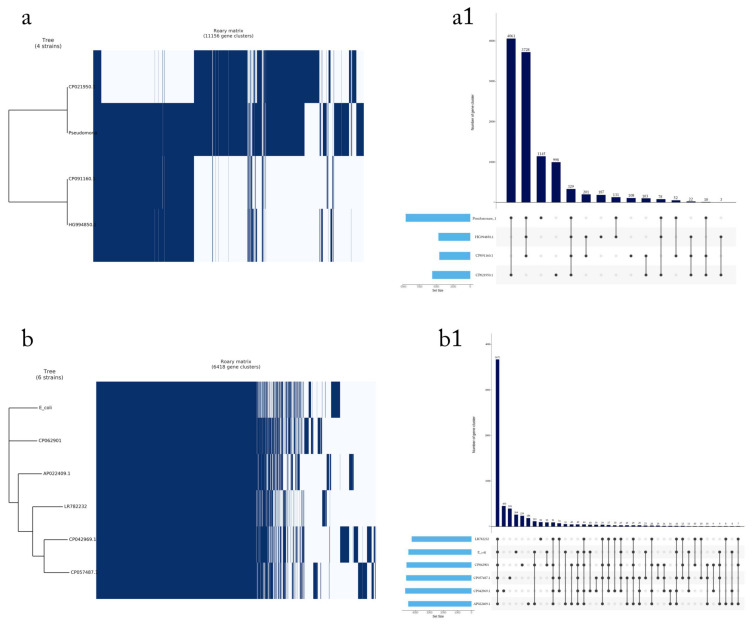
Pan-genome analysis of (**a**) gene clusters (total = 11,156), shell genes (10,826), and core genes (330) for *Pseudomonas*_1. Gene presence/absence of 4061 gene clusters commonly present in four strains (**a1**)**.** (**b**) Gene clusters (total = 6418), shell genes (2746), and core genes (3672) for *E. coli*_7. Gene presence/absence of 3671 gene clusters commonly present in six strains (**b1**). (**c**) Gene clusters (total = 11,848), shell genes (11,523), and core genes (325) for *K. pneumoniae*_3. Gene presence/absence of 324 gene clusters commonly present in the six strains (**c1**). (**d**) Gene clusters (total = 6003), shell genes (1076), and core genes (4927) for *K. pneumoniae*_3. Gene presence/absence of 4926 gene clusters commonly present in the six strains (**d1**). (**e**) Gene clusters (total = 3249), shell genes (1097), and core genes (2152) for *S. aureus*_9. Gene presence/absence of 2151 gene clusters commonly present in the four strains (**e1**). (**f**) Gene clusters (total = 2763), shell genes (370), and core genes (2393) for *S. aureus*_10. Gene presence/absence of 2392 gene clusters commonly present in the four strains (**f1**). (**g**) Gene clusters (total = 3022), shell genes (740), and core genes (2282) for *S. aureus*_12. Gene presence/absence of 2281 gene clusters commonly present in the four strains (**g1**).

**Figure 5 microorganisms-11-02432-f005:**
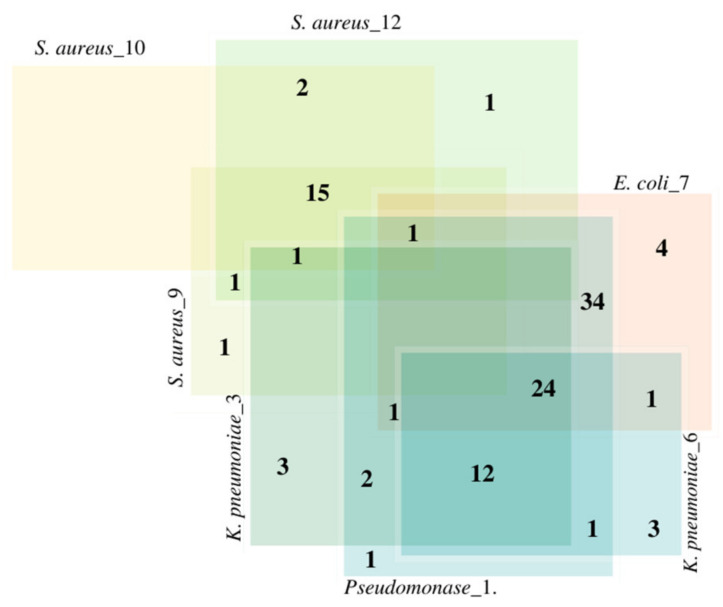
Venn diagram showing the susceptibility of multidrug-resistant bacteria isolated from selected hospitals in Jeddah to 108 universal antibiotics (data on antibiotics are available in Appendix A).

**Table 1 microorganisms-11-02432-t001:** The index information of the isolated bacteria.

Sample ID	Index 5	Index 7
Pseudomonas_1	CTAGGCAA	CTCACCAA
K_pneumoniae_3	CTGTATTA	ATATGGAT
K_pneumoniae_6	TCACGCCG	GCGCAAGC
E_coli_7	ACTTACAT	AAGATACT
S_aureus_9	GTCCGTGC	GGAGCGTC
S_aureus_10	AAGGTACC	ATGGCATG
S_aureus_12	GGAACGTT	GCAATGCA

## Data Availability

Data of biochemical identification of MDR bacteria are available in the Ministry of Health Jeddah branch. Data of molecular identification are available on Blast; BLAST: Basic Local Alignment Search Tool (nih.gov) (accessed on 22 May 2022).

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
