# Peer review of "Comparative De Novo and Pan-Genome Analysis of MDR Nosocomial Bacteria Isolated from Hospitals in Jeddah, Saudi Arabia"

_microorganisms, 2023, doi:10.3390/microorganisms11102432_

Round 1
Reviewer 1 Report (Previous Reviewer 4)
Comments
Material and methods:
How many isolates have you sequenced? In the results you mentioned it was 7 isolates bud it needs to be mentioned here too. It is not clear from methods. How many bacteria were analysisd for phylogenetic tree and how many were selected for the WGS? Were the bacteria isolates the same for phylogenetic analysis and WGS?
2.4. 16S rRNA Gene Sequence and Phylogenetic Analysis:
What exact type of Illumina sequencer did you use for the sequencing? How long (how many bp) were the reads obtained after sequencing? Still missing in this chapter.
7 MDR bacteria were sequenced for 16S analysis – why and which strains exactly?
2.5. WGS and Analysis Workflow
You are describing here the general workflow of Ilumina seq.platform – no need. What exact type of Ilumina sequencer did you use for the sequencing?
2.6. De novo assembly
You mentioned NovaSeq 6000 seq. machine – this should be mentioned in previous chapter.
What was the generated length of reads?
“(some things are missing here?)” – what is it???
2.7. Pan genome analysis
Still not getting why you wanted to identify taxonomic boundaries of the genomes?
3. Statistical analysis:
The phylogenetic indices… table 1 – what is it? Index 5 and Index 7? What is the table saying?
Results:
4.1. Identification of isolated bacteria
You are saying you picked 7 isolates for further analysis: 2 E. coli, 2 Klebsiella sp. and 3 Staphylococcus sp. But then in the following chapter – the 16S analysis - there is mentioned Pseudomonas aeruginosa (PSAE) – why?
Also, the results for PSAE from 16S seq. differed from previous tests – how? What was the result of 16S for this PSAE isolate?
What was the point of this analysis? The images are unreadable. The same goes for all pictures in this article.
4.3. Phylogenetic tree of the seven MDR bacteria
Why did you analysis the taxonomic boundaries of different bacterial species?
What was the point of creating the phylogenetic tree for different bacterial species based on individual contigs? Why were the particular contigs used for this? What were you trying to prove by this analysis?
When characterizing isolates of MDR bacteria, it is useful to know what sequence type they are – when you have data from WGS, should not be a problem to find out.
4.3. MDR genes
Table containing info about which resistance genes at bacterial species were detected is missing. What type of OXA, CTX-M enzymes exactly?
All image captions are not readable. So many figures are not appropriate, you should process the data from the figures and put them in tables.
Unfortunately, I cannot recommend the article for acceptance.
Author Response
Dear Reviewer1,
Kindly, find below our response to your comments:
Comments
Material and methods:
How many isolates have you sequenced? In the results you mentioned it was 7 isolates bud it needs to be mentioned here too. It is not clear from methods. How many bacteria were analysisd for phylogenetic tree and how many were selected for the WGS? Were the bacteria isolates the same for phylogenetic analysis and WGS?
Thank you for your comment. Number of isolates were added in methods.
Yes, the 7 isolated MDR bacteria were analysed for the phylogenetic tree, Denovo, and WGS.
2.4. 16S rRNA Gene Sequence and Phylogenetic Analysis:
What exact type of Illumina sequencer did you use for the sequencing? How long (how many bp) were the reads obtained after sequencing? Still missing in this chapter.
The 16S rRNA was done at KFMC (KAU) and the information was added to methods section and highlighted.
7 MDR bacteria were sequenced for 16S analysis – why and which strains exactly?
Thank you for your comment, the answers of these two questions were removed according to reviewer 2 and 4 recommendation in the first submission, as it is result not methods.
2.5. WGS and Analysis Workflow
You are describing here the general workflow of Ilumina seq.platform – no need. What exact type of Ilumina sequencer did you use for the sequencing?
Thank you for your comment. The type of Ilumina sequencer was added to the method section and highlighted.
2.6. De novo assembly
You mentioned NovaSeq 6000 seq. machine – this should be mentioned in previous chapter.
Thank you for your suggestion, the previous chapter was revised.
What was the generated length of reads?
“(some things are missing here?)” – what is it???
Thank you for question, the generated length of reads were added. The method of De novo sequencing was done in Macrogen, and the analysis report and QC report were added as supplementary to the manuscript.
2.7. Pan genome analysis
Still not getting why you wanted to identify taxonomic boundaries of the genomes?
Thees isolates were isolated from hospitals in Jeddah, Saudi Arabia, to know and understand these isolates we need to compare them by the known MDR bacteria in the world, and if there are differences between them. Also, this will register these new isolates in the gene bank.
- Statistical analysis:
The phylogenetic indices… table 1 – what is it? Index 5 and Index 7? What is the table saying?
This table was added according to reviewer’s suggestion in the first submission, we can remove it if you suggest that.
Results:
4.1. Identification of isolated bacteria
You are saying you picked 7 isolates for further analysis: 2 E. coli, 2 Klebsiella sp. and 3 Staphylococcus sp. But then in the following chapter – the 16S analysis - there is mentioned Pseudomonas aeruginosa (PSAE) – why?
Thank you for your comment. Yes, the first identification by biochemical reaction, this isolate was identified as Pseudomonas aeruginosa, so, we named it Pseudomonas_1, and sent the DNA extract to Macrogen with this mane, but the molecular identification showed it’s not Pseudomonas, this isolate did not identify at King Fahad medical research centre (KFMRC). The identification diagram showed the percentage of taxonomic species, figure 1a. We couldn’t change the name from the original images and the supplementary files. The scientific editor in enago suggest keeping the text as it shown. If you recommend something ells, I will be happy to do it.
Also, the results for PSAE from 16S seq. differed from previous tests – how? What was the result of 16S for this PSAE isolate?
Kindly, would you mention the difference of which text by adding the sentence line? The results of 16S for the isolate PSAE was E. coli since it has the highest match rate with isolates in BLAST.
What was the point of this analysis? The images are unreadable. The same goes for all pictures in this article.
What analysis? All the images were made by Macrogen, and they send them as images with the resolution consistent with the journal’s instructions (“a minimum resolution of 600 dpi”).
4.3. Phylogenetic tree of the seven MDR bacteria
Why did you analysis the taxonomic boundaries of different bacterial species?
We analysis the taxonomic boundaries of the seven bacterial species to know if our isolates are new strains or it were imported to our country.
What was the point of creating the phylogenetic tree for different bacterial species based on individual contigs? Why were the particular contigs used for this? What were you trying to prove by this analysis?
As I said in the previous answer to know if our isolates are new strains, or it were imported to our country.
When characterizing isolates of MDR bacteria, it is useful to know what sequence type they are – when you have data from WGS, should not be a problem to find out.
Thank you for your information.
4.3. MDR genes
Table containing info about which resistance genes at bacterial species were detected is missing. What type of OXA, CTX-M enzymes exactly?
Thank you for your comment. Tables of tested resistance genes at bacterial species, we prefer to add the figures in the article, and the tables were added as supplementary tables due to the long table data. We had mention this in the result section, and we revised it and added the tables number in the supplementary “revised text is highlighted”.
About the type of OXA, CTX-M enzymes, I am not sure what you mean with your question? These two genes are part of Beta-lactamase enzyme, but from two classes: from class A CTX-M beta-lactamase, and OXA beta-lactamase from class D. Is this the answer you looking for?
All image captions are not readable. So many figures are not appropriate, you should process the data from the figures and put them in tables.
All images were provided from Macrogen as JPG, we did not capture them, but to arrange the figures and title each image inside the figure we did that by Canvas.com, and download them as PNG, and we used high resolution and tech them if they within the journal required.
Unfortunately, I cannot recommend the article for acceptance.
I respect your opinion, but I have the right to know why you cannot recommend the article for acceptance? This is our second submission, the first submission three from four reviewers accept the article, and Editor gave reason for rejection the geographical limitation, which is not acceptable due to many reasons I present them in response to the Editor. Anyway, ethically there are guided to evaluate any manuscript, and as scientific researchers we need to follow these guidelines, and I believe that all of you do that.
Reviewer 2 Report (New Reviewer)
I have read with interest the mansucript, considering the fact that AMR represents a major concearn worlwide. I have some comments to be addressed in order to improve the quality of the manuscript:
Abstract:
"Staphylococcus aureus (MRSA)" - the whole meaning of MRSA should be described.
pay attention the punctuation marks
Introduction
why do you refer only to KPC and not the other carbapenemases?
How about the mechanics of resistance to other antibiotic classes (such as fluoroquinolones or AG)?
There seem to be multiple pieces of information provided, without the source mentioned.
Enterobacteriaceae -> Enterobacterales
"This study aims to identify MDR isolates based on the definition of MDR bacteria as those that are resistant to at least three or more antibiotic classes"- how can a microorganism be resistant to an antibiotic class?
MDR is defined as acquired non-susceptibility to at least one agent in three or more antimicrobial categories
Results:
It is not mentioned that the infections are hospital-acquired (nosocomial). How did you determine this fact?
I highly recommend changing the format of the pictures, due to the fact that the text is not understandable at all. The legend for figures 1a,b,c seems wrong.
The authours should consider includding the antibiotic resistance rates to each antibiotic tested, especially considering the fact that they included this aspect in the discussion section. Also, consider adding some information about the site of infection, which pathogen is more common in certain infections, etc.
Discussion
Again, multiple pieces of information are without references, especially in the beginning.
There is multiple general information that should better be included in the introduction section.
Consider rephrasing the discutions and conclusions, to emphasize on the particularity of the study and the novelty provided by your findings. What is the result of your comparisom?
minor edit required.
Author Response
Dear reviewer2,
Kindly, find below our response to your comments:
Comments and Suggestions for Authors
I have read with interest the mansucript, considering the fact that AMR represents a major concearn worlwide. I have some comments to be addressed in order to improve the quality of the manuscript:
Abstract:
"Staphylococcus aureus (MRSA)" - the whole meaning of MRSA should be described. Done
pay attention the punctuation marks
Introduction
Why do you refer only to KPC and not the other carbapenemases?
Thank you for your commitment, we refer to KPC as example of the function of the four groups of carbapenemases. We are giving a brief introduction for the aim of the research article. if We elaborated example from each group of carbapenemases the introduction will become review article. I respect your opinion, and revise this section within the content.
How about the mechanics of resistance to other antibiotic classes (such as fluoroquinolones or AG)?
We refer to the most documented mechanisms of resistance to antibiotics as example, and we think this is enough in research article so we can keep the reader’s interest.
There seem to be multiple pieces of information provided, without the source mentioned.
Please, can you highlight these multiple pieces of information?
Enterobacteriaceae -> Enterobacterales
"This study aims to identify MDR isolates based on the definition of MDR bacteria as those that are resistant to at least three or more antibiotic classes"- how can a microorganism be resistant to an antibiotic class?
MDR is defined as acquired non-susceptibility to at least one agent in three or more antimicrobial categories
Thank you for your comment. Yes, the word class is used to categorize antibiotics. This can be found in CDC Glossary of Terms Related to Antibiotic Resistance:
Glossary of Terms Related to Antibiotic Resistance | NARMS | CDC
Results:
It is not mentioned that the infections are hospital-acquired (nosocomial). How did you determine this fact?
Thank you for your query. The isolated bacteria have been isolated from patient and awards, then they have been identified chemically and the antibiotics resistance and sensitivity were tested. This is described in 2.2 Clinical Specimens.
I highly recommend changing the format of the pictures, due to the fact that the text is not understandable at all. The legend for figures 1a,b,c seems wrong.
What do you mean with changing the format of the pictures? And what is the relationship between the text and the format of the pictures? The figures are used to show the results of the sequences and all the figures were made by Macrogen, and they send them as images with the resolution consistent with the journal’s instructions (“a minimum resolution of 600 dpi”).
Please, pinpointing which part in the results is not understandable?
The authours should consider includding the antibiotic resistance rates to each antibiotic tested, especially considering the fact that they included this aspect in the discussion section. Also, consider adding some information about the site of infection, which pathogen is more common in certain infections, etc.
Thank you for your suggestions, the results of antibiotics test are under editing for another publication, the results are considered as antibiogram study. This study has molecular investigations for species of MDR bacteria, and resistant genes.
Discussion
Again, multiple pieces of information are without references, especially in the beginning.
I would appreciate it if you gave the line number to your comment, this will be helpful!
I checked the discussion point by point and I couldn’t find any uncited text. The first paragraph is not cited paragraph, we just added brief before starting the discussion.
There is multiple general information that should better be included in the introduction section.
Please could you send the line number for which part you recommend moving to the introduction.
Consider rephrasing the discutions and conclusions, to emphasize on the particularity of the study and the novelty provided by your findings. What is the result of your comparisom?
Conclusion was revised.
Round 2
Reviewer 1 Report (Previous Reviewer 4)
First of all, I thank the authors of the manuscript for editing the text based on my comments.
The text has been adequately edited and I am now pleased to recommend the manuscript for acceptance.
Author Response
Thank you so much for your time and descension.
Reviewer 2 Report (New Reviewer)
As per MDPI instructions, ALL changes performed to the manuscript should be marked, preferably using track changes. It is extremely hard for a reviewer to compare the original and revised manuscript, especially since a considerable period of time has passed since the initial report. You only highlighted some punctual changes. Now, I have to read again all the information previously reviewed, because I can not know what you changed.
- Glossary of Terms Related to Antibiotic Resistance | NARMS | CDC - MDR -CDC typically uses this term to refer to an isolate that is resistant to at least one antibiotic in three or more drug classes.
As I mentioned before, the microorganism can be resistant to an antibiotic from a class of atb, not to a class of antibiotics.
Thank you for your query. The isolated bacteria have been isolated from patient and awards, then they have been identified chemically and the antibiotics resistance and sensitivity were tested. This is described in 2.2 Clinical Specimens.
Not all bacteria isolated from hospitalized patients determine nosocomial infections. If the microorganism was isolated at addition, there is are high chance for it to be Community-acquired. T The definition of nosocomial should be included, especially since you used this term in the title.
What do you mean with changing the format of the pictures? And what is the relationship between the text and the format of the pictures?
I was referring to the text included in the figures. Even if the resolution of the pictures is adequate, the reader has to zoom in at least 400% in order to understand something; this might decrease the reader's interest.
I would appreciate it if you gave the line number to your comment, this will be helpful!
For example, rows 352-361 don t have any source
Please could you send the line number for which part you recommend moving to the introduction.
For example, 373-384, 420-433.
Consider rephrasing the discussions and conclusions, to emphasize the particularity of the study and the novelty provided by your findings. What is the result of your comparison?
minor changes required.
Author Response
Dear reviewer,
First, I would like to express my regret for this issue. I agree with the reviewer’s comment, and I highlighted the changes in yellow.
- Glossary of Terms Related to Antibiotic Resistance | NARMS | CDC - MDR -CDC typically uses this term to refer to an isolate that is resistant to at least one antibiotic in three or more drug classes.
As I mentioned before, the microorganism can be resistant to an antibiotic from a class of atb, not to a class of antibiotics.
Thank you for your comment, but I did not get your point. Previously, you have mentioned that ‘MDR is defined as acquired non-susceptibility to at least one agent in three or more antimicrobial categories’ and I am not sure what you mean by atb. As I know ATB is the abbreviation of antibiotics, the following definitions of atb are obtained from three websites:
ATB in Medical, Meanings and Abbreviations (acronym24.com)
ATB - What does ATB Stand For in Medical & Science ? (acronymsandslang.com)
ATB Healthcare Abbreviation Meaning (allacronyms.com)
We have also performed antimicrobial susceptibility testing to each isolate of not less than 21 antibiotics belonging to more than 10 antibiotic classes, including beta-lactam antibiotics, penicillin, aminopenicillin, cephalosporin and all generations of cephalosporin (1–4), amphenicol, aminoglycoside, streptogramin, monobactams, quinolone, carbapenem, glycylcyclines and tetracycline. These antibiotics are accredited by the Ministry of Health in Saudi Arabia. The classes of antibiotics can be checked from the following link:
All Antibiotic Classes | A.R. & Patient Safety Portal (cdc.gov)
Thank you for your query. The isolated bacteria have been isolated from patient and awards, then they have been identified chemically and the antibiotics resistance and sensitivity were tested. This is described in 2.2 Clinical Specimens.
Not all bacteria isolated from hospitalized patients determine nosocomial infections. If the microorganism was isolated at addition, there is are high chance for it to be Community-acquired. T The definition of nosocomial should be included, especially since you used this term in the title.
Thank you for your comment. It is correct that some isolated bacteria are not nosocomial and such bacteria may be Community-acquired, but one of the isolate selection conditions is the frequency of these isolates from several wards in the hospitals.
With regard to the second part of your comment, ‘the definition of nosocomial should be included, especially since you used this term in the title’. We can add the definition of nosocomial to the manuscript, but, as a professor teaching medical microbiology, we taught students this term in undergraduate courses. Thus, adding such a term in a research article that aims to isolate nosocomial MDR and genes detected from these MDR isolates will be information already known to readers. The definition of nosocomial is added and highlighted.
What do you mean with changing the format of the pictures? And what is the relationship between the text and the format of the pictures?
I was referring to the text included in the figures. Even if the resolution of the pictures is adequate, the reader has to zoom in at least 400% in order to understand something; this might decrease the reader's interest.
Thank you for your clarification. As I said previously, the images were made by Macrogen and they were sent as images and in a pdf report. We have also checked the resolution of the images to ensure that they are compatible with the journal instructions. At this time, I sent these images to a specialist in diagram to clear the numbers and letters. If these images are okay, then the journal can use them.
I would appreciate it if you gave the line number to your comment, this will be helpful!
For example, rows 352-361 don t have any source
These lines are an introduction that I wrote by myself and are not quoted from a source.
Please could you send the line number for which part you recommend moving to the introduction.
For example, 373-384, 420-433.
Thank you for your comment. Given the arrangement of references, the text was modified to fit the discussion section. I hope this fix the issue.
Consider rephrasing the discussions and conclusions, to emphasize the particularity of the study and the novelty provided by your findings. What is the result of your comparison?
The discussions and conclusions were revised.

This manuscript is a resubmission of an earlier submission. The following is a list of the peer review reports and author responses from that submission.
Round 1
Reviewer 1 Report
The manuscript aims to perform a comparative analysis of MDR strains isolated from hospitals in Saudi Arabia. In its current form, the manuscript is not acceptable and needs to be reorganized and rewritten.
My major concerns are:
1. The methodology section is vague and unprofessionally written. Key pieces of information essential for validation of the results including methodology for sample collection, isolation and cultivation of strains or sequencing platform used is missing. Still, authors prefer to invest a lot of text in presenting what is basically an excerpt from the Qiagen DNA extraction kit manual or some general text regarding the Illumina SBS technology. NGS data processing and bioinformatics analysis are especially poorly described. Without proper technical details in the methodology section, the whole study is unreproducible and untrustful.
2. The NGS data presented here should be deposited to the appropriate databases: Illumina short-reads to NCBI SRA and the draft genomes (contigs) of the isolated strains to NCBI Genome
3. Figures are mislabeled. Figures 1(pg4) and 2(pg5) are very general and bring no novelty to the study. They might be used by the authors to properly organize their bioinformatics-related methodology section, but by no means can be considered as independent figures with relevance for this study. Figure 1(pg7), Figure 3 and Figure 4 are of low quality and hence, unreadable.
4. Discussions are very general and very seldom make reference to the data presented here.
5. Without proper details in the text, the title is misleading. Apparently, samples were from a single city, and hence can not be representative of Saudi Arabia. The number of hospitals from where the samples were collected is vaguely presented as several. What kind of hospitals and samples are we talking about: adults vs kids, male vs females, ill vs healthy?
Author Response
My major concerns are:
- The methodology section is vague and unprofessionally written. Key pieces of information essential for validation of the results including methodology for sample collection, isolation and cultivation of strains or sequencing platform used is missing. Still, authors prefer to invest a lot of text in presenting what is basically an excerpt from the Qiagen DNA extraction kit manual or some general text regarding the Illumina SBS technology. NGS data processing and bioinformatics analysis are especially poorly described. Without proper technical details in the methodology section, the whole study is unreproducible and untrustful.
Thank you for these valuable comments, which have helped us to improve the methodology section. The
- The NGS data presented here should be deposited to the appropriate databases: Illumina short-reads to NCBI SRA and the draft genomes (contigs) of the isolated strains to NCBI Genome
Thank you for the comment; the contigs were added in the Materials and Methods section.
- Figures are mislabeled. Figures 1(pg4) and 2(pg5) are very general and bring no novelty to the study. They might be used by the authors to properly organize their bioinformatics-related methodology section, but by no means can be considered as independent figures with relevance for this study. Figure 1(pg7), Figure 3 and Figure 4 are of low quality and hence, unreadable.
Thank you for pointing out these shortcomings. Figures 1 and 2 are by Macrogen and show the protocol used for each analysis. Our analysis of bacterial isolates detected the resistance genes, and we can remove them if doing so seems preferable.
Figures 1, 3, and 4 are images provided by Macrogen; the resolution for all of them figures exceeds 600 dpi, consistent with the journal’s instructions (“a minimum resolution of 600 dpi”).
- Discussions are very general and very seldom make reference to the data presented here.
Thank you for the valuable suggestion, which has helped to improve the discussion.
- Without proper details in the text, the title is misleading. Apparently, samples were from a single city, and hence can not be representative of Saudi Arabia. The number of hospitals from where the samples were collected is vaguely presented as several. What kind of hospitals and samples are we talking about: adults vs kids, male vs females, ill vs healthy?
Thank you for these comments; the title has been modified as suggested, and the number and the type of hospitals are now specified in the Materials and Methods section, indicating that we compared adults and children, men and women, and ill and healthy individuals.
Reviewer 2 Report
The title mentioned “Saudi Arabia” but the included hospitals were in one city. Please change the title to mention “Jeddah, Saudi Arabia”.
Abstract: The conclusion was not appropriate and is not supported by this study. The authors only looked at MDR & not all bacteria. Please revise carefully.
It seems there are some mistakes with references. Please use the original and correct references. For example:
Ref 12 seems to be incorrect (tuberculosis!)
Ref 15 & 14. The 3 hospitals were ref 14 in the text but ref 15 in the reference list.
Methods:
I believe you need ethical approval. These bacteria are from samples taken from human subjects.
The study included samples from some random units (excluded medical & surgical floors) & included both pediatrics and adults, which is not recommended. At least mention this as a limitation as adults & those in ICUs tend to have more MDR bacteria.
Statistical analysis: please the part about p<0.01 as highly significant. This is not acceptable statistically. <0.05 as significant is sufficient.
Author Response
The title mentioned “Saudi Arabia” but the included hospitals were in one city. Please change the title to mention “Jeddah, Saudi Arabia”.
Done.
Abstract: The conclusion was not appropriate and is not supported by this study. The authors only looked at MDR & not all bacteria. Please revise carefully.
Done.
It seems there are some mistakes with references. Please use the original and correct references. For example:
Ref 12 seems to be incorrect (tuberculosis!)
Ref 15 & 14. The 3 hospitals were ref 14 in the text but ref 15 in the reference list. Done
Methods:
I believe you need ethical approval. These bacteria are from samples taken from human subjects.
Thank you for these valuable observations. We concur; the parts of our study involving human subjects indeed received ethical approval. We recorded the file number of each patient, and these data are available in the hospital’s database. The relevant wording is “this is mentioned in each publication or thesis with such human subjects,” but, in the case of this study (which is part of my master’s thesis), the proposal was approved without the involvement of the IRB. Please click on the link below for more details about such exceptions to IRB approval.
(b) Unless otherwise required by department or agency heads, research activities in which the only involvement of human subjects will be in one or more of the following categories are exempt from this policy:
(4) Research involving the collection or study of existing data, documents, records, pathological specimens, or diagnostic specimens, if these sources are publicly available or if the information is recorded by the investigator in such a manner that subjects cannot be identified, directly or through identifiers linked to the subjects.
https://www.hhs.gov/ohrp/regulations-and-policy/regulations/regulatory-text/index.html#46.101The study included samples from some random units (excluded medical & surgical floors) & included both pediatrics and adults, which is not recommended. At least mention this as a limitation as adults & those in ICUs tend to have more MDR bacteria.
Thank you again for the valuable comments. Yes, we took random samples from several places, including floors, walls, equipment, and furniture, from most of the hospital wards (e.g., the ER, OPD, ICU, kidney clinic, NICU, OB, OR, and staff areas. Further, the clinical specimens were taken from the hospital laboratory with consideration of the gender and age of the patients, the sources of the specimen, and the wards involved.
Statistical analysis: please the part about p<0.01 as highly significant. This is not acceptable statistically. <0.05 as significant is sufficient. Done
Reviewer 3 Report
The authors communicate on the multidrug-resistance bacteria isolated from hospitals in Saudi Arabia. Indeed there is little information on this and thus making it of interest. However, the results are not shown clearly, which makes the reader have to go back and read again, and some results are presented in the abstract and conclusion, but not in the other sections. I couldn't analyze the figures because they are too small and blur. Here are some points for corrections and suggestions:
Abstract:
Please, add the number of isolates before the percentage, correct it throughout the text. Example: 31 (55.0%)
‘There were a total of 108 isolates, 75 of which were MDR bacteria’ add the percentage 75 (X%) and replace were for was.
Introduction
“…More alarming is the presence of MDR bacteria, which refer to bacteria...” – replace MDR for multidrug-resistant (MDR). After that, you can use only MDR throughout the text
“…species of the Enterobacteriaceae family was the most frequently isolated, with E. coli being the most common organism at 40.0%.” replace was for were and “at 40.0%” for representing 40.0% of the total.
Material and Methods
2.2. Clinical Specimens:
In the abstract you say there were 108 isolates, add this information in the Material and Methods. Of the 108 isolates, 75 were MDR? Add this information as well.
Specify the period the isolates were collected
How the bacteria strains were isolated? Is there a previous paper that used these same strains and describe how they were isolated so you can cite it?
2.3. Isolation of Multidrug-Resistant (MDR) Bacteria:
Of 108 isolates, 75 (X%) were MDR…
Please, add the number of isolates and the percentage they represent among the number of MDR isolates for all bacteria species you cited, which are S. aureus (MRSA), multidrug-resistant S. epidermidis, multi-drug-resistant S. pneumoniae, ESBL-producing Gram-negative Enterobacteriaceae such as ESBL E. coli and ESBL K. pneumoniae, E. cloacae, and multidrug-resistant P. aeruginosa
“The MDR bacteria E. coli, K. pneumoniae, and S. aureus were selected for the following analyses”. How many isolates of each bacteria species were selected for the analyses? In the conclusion you refer to seven isolates that were analyzed, but you haven’t mention them clearly in Material and Methods and Results section.
2.6. Sequencing and Analysis Workflow
“Sequencing: For library construction, DNA was extracted from the isolated bacteria according to an appropriate protocol, depending on the 7 isolated bacteria”. Please, specify what are these seven isolates. You need to make clear the number of isolates of each bacteria species in the Material and Methos section.
Small numbers ranging from one to ten should generally be spelled out. Larger numbers (above ten) are written as numerals. Please, replace 7 for seven
Results:
Make a table with the resistance genes found in the MDR isolates. You can do it by genera of Gram-negative (e.g. Klebsiella, Escherichia, Citrobacter) and Gram-positive (e.g. staphylococcus) instead of species to make the table shorter.
You need to clarify how many bacteria isolates you have in total, how many isolates are MDR, how many isolates were selected for further analysis and had the resistance genes analyzed. In the abstract you said ‘There were a total of 108 isolates, 75 of which were MDR bacteria.’ and in Figure 5 it’s written ‘Venn diagram of the susceptibility of 108 multidrug-resistant bacterial isolates from selected hospitals in Jeddah to universal antibiotics’. You need to write the correct information through the manuscript.
How many isolates were identified according to 16S rRNA? Add numbers before the percentage, the data is confusing.
Discussion
You need to improve the discussion. You discussed about tet gene, but not even mention the resistant genes detected in the MDR isolates of your work. You need to discuss about SHV, OXA, CTX-M, TEM-1, NDM-1, VIM-1, ermA, ermB, ermC, msrA, qacA, qacB and qacC
You have to describe the study you used to compare your results.
Multidrug-resistance is defined as resistance to antimicrobial drugs from at least three or more categories. Please, check if the definition is correct throughout the text.
Conclusion
Please, rewrite the conclusion to make it clearly.
Check for punctuations, sometimes a period is better than a comma.
Delete the following sentence ‘In this study, seven strains of bacteria belonging to four genera were isolated, and the results showed that these bacteria are resistant to more than three anti-biotics. Genetic analysis was done to identify antibiotic resistance genes’. You are repeating Results and Material and Methods in the conclusion.
Replace ‘The need to identify the types’ for ‘There is a need to identify the types’
Replace ‘multi drugs resistant’ for ‘multidrug-resistant’
Use hyphen in Gram-negative and Gram-positive
Author Response
Comments and Suggestions for Authors
The authors communicate on the multidrug-resistance bacteria isolated from hospitals in Saudi Arabia. Indeed there is little information on this and thus making it of interest. However, the results are not shown clearly, which makes the reader have to go back and read again, and some results are presented in the abstract and conclusion, but not in the other sections. I couldn't analyze the figures because they are too small and blur. Here are some points for corrections and suggestions:
Abstract:
Please, add the number of isolates before the percentage, correct it throughout the text. Example: 31 (55.0%)
‘There were a total of 108 isolates, 75 of which were MDR bacteria’ add the percentage 75 (X%) and replace were for was.
Thank you; we have made these changes.
Introduction
“…More alarming is the presence of MDR bacteria, which refer to bacteria...” – replace MDR for multidrug-resistant (MDR). After that, you can use only MDR throughout the text
“…species of the Enterobacteriaceae family was the most frequently isolated, with E. coli being the most common organism at 40.0%.” replace was for were and “at 40.0%” for representing 40.0% of the total.
Thank you; we have made these changes.
Material and Methods
2.2. Clinical Specimens:
In the abstract you say there were 108 isolates, add this information in the Material and Methods. Of the 108 isolates, 75 were MDR? Add this information as well.
Done.
Specify the period the isolates were collected
Done.
How the bacteria strains were isolated? Is there a previous paper that used these same strains and describe how they were isolated so you can cite it?
Thank you for pointing out this omission. We explain that the bacteria strains were isolated from the hospital laboratory and that we received a blood agar Petri dish of each pure culture.
2.3. Isolation of Multidrug-Resistant (MDR) Bacteria:
Of 108 isolates, 75 (X%) were MDR…
Thank you; we have made these changes.
Please, add the number of isolates and the percentage they represent among the number of MDR isolates for all bacteria species you cited, which are S. aureus (MRSA), multidrug-resistant S. epidermidis, multi-drug-resistant S. pneumoniae, ESBL-producing Gram-negative Enterobacteriaceae such as ESBL E. coli and ESBL K. pneumoniae, E. cloacae, and multidrug-resistant P. aeruginosa
Thank you; we have made these changes.
“The MDR bacteria E. coli, K. pneumoniae, and S. aureus were selected for the following analyses”. How many isolates of each bacteria species were selected for the analyses? In the conclusion you refer to seven isolates that were analyzed, but you haven’t mention them clearly in Material and Methods and Results section.
Thank you; we have made these changes.
2.6. Sequencing and Analysis Workflow
“Sequencing: For library construction, DNA was extracted from the isolated bacteria according to an appropriate protocol, depending on the 7 isolated bacteria”. Please, specify what are these seven isolates. You need to make clear the number of isolates of each bacteria species in the Material and Methos section.
Small numbers ranging from one to ten should generally be spelled out. Larger numbers (above ten) are written as numerals. Please, replace 7 for seven
Thank you; we have made these changes.
The seven isolates are specified in the revised version of section 2.3, Isolation of multidrug-resistant (MDR) bacteria
Results:
Make a table with the resistance genes found in the MDR isolates. You can do it by genera of Gram-negative (e.g. Klebsiella, Escherichia, Citrobacter) and Gram-positive (e.g. staphylococcus) instead of species to make the table shorter.
In light of this valuable comment, we have made WGS to show the presence and absence of resistant genes to 108 antibiotics. We are submitting supplementary tables to present these results (Tables S1-S7 and the antibiotics tested in Table S16). Other results were added to Figure 4 since they require less space.
You need to clarify how many bacteria isolates you have in total, how many isolates are MDR, how many isolates were selected for further analysis and had the resistance genes analyzed. In the abstract you said ‘There were a total of 108 isolates, 75 of which were MDR bacteria.’ and in Figure 5 it’s written ‘Venn diagram of the susceptibility of 108 multidrug-resistant bacterial isolates from selected hospitals in Jeddah to universal antibiotics’. You need to write the correct information through the manuscript.
Thank you for these valuable comments. There 108 total isolates, and we selected 7 of the 75 MDR isolates, of which more than 20% these were E. coli (2), K. pneumonia (2), or S. aureus (3).
In Figure 5, the figure of 108 is the numbers of antibiotics tested against our isolates using Macrogen. We apologize for the typographical error; the number was corrected as the number of tested antibiotics.
We tested the isolates against the 21 antibiotics approved by the Saudi Ministry of Health, and the results showed the MDR isolates that were resistant to three or more antibiotics.
How many isolates were identified according to 16S rRNA? Add numbers before the percentage, the data is confusing.
The total number of isolates identified according to 16S rRNA was 18, of which we chose 7 for further study.
Discussion
You need to improve the discussion. You discussed about tet gene, but not even mention the resistant genes detected in the MDR isolates of your work. You need to discuss about SHV, OXA, CTX-M, TEM-1, NDM-1, VIM-1, ermA, ermB, ermC, msrA, qacA, qacB and qacC
You have to describe the study you used to compare your results.
Thank you for your the suggestion, which has helped to improve the discussion.
Multidrug resistance is defined as resistance to antimicrobial drugs from at least three or more categories. Please, check if the definition is correct throughout the text.
Thank you for pointing this out; we checked the text to ensure that the definition of MDR was correct throughout the text.
Conclusion
Please, rewrite the conclusion to make it clearly.
Check for punctuations, sometimes a period is better than a comma.
Delete the following sentence ‘In this study, seven strains of bacteria belonging to four genera were isolated, and the results showed that these bacteria are resistant to more than three anti-biotics. Genetic analysis was done to identify antibiotic resistance genes’. You are repeating Results and Material and Methods in the conclusion.
Done.
Replace ‘The need to identify the types’ for ‘There is a need to identify the types’ Done.
Replace ‘multi drugs resistant’ for ‘multidrug-resistant’ Done.
Use hyphen in Gram-negative and Gram-positive Done.
Reviewer 4 Report
Introduction
Rewrite the whole section – it is not necessary to describe the general mechanism of action of beta-lactam antibiotics and general resistance mechanisms + citations are shifted:
„Regarding the Gram-positive MDR bacteria, a study conducted on the incidence of Gram-positive MDR bacteria at King Abdulaziz University Hospital (KAUH), Jeddah, found that the most prevalent isolates were S. aureus (MRSA), at 20% [15]“
Check the citations and reference list, it appears that citations are somehow shifted. It occurs to me that instead of [15]“there should be [17]“ etc.
Materials and methods
2.2 Clinical specimens
How many isolates did you have in total – in abstract, you mentioned 108 – it needs to be mentioned also here. How many have you sequenced? 75 or 7? It is not clear from methods. How many bacteria were analysed for phylogenetic tree and for the resistance determinants analysis? How did you determine the susceptibility of isolates? You mentioned Phoenix BD system in abstract but there is no information here, in methods.
2.3. Isolation of Multidrug-Resistant (MDR) Bacteria
Should be mentioned in results.
2.4. 16S rRNA Gene Sequence and Phylogenetic Analysis
1. Correct the citations, e.g.:
„Bacterial DNA was extracted using the Qiagen DNA extract kit as per manufactur-er’s instructions [16].“
[18] instead of [16]
„The isolated DNA samples were stored at −20 °C according to the manufacturer’s protocol, as outlined in the DNeasy Blood and Tissue Handbook [17].“
[18] instead of [17]...or it is not necessary to mention this handbook again – generally is extracted DNA kept in -20.
„The primers for the amplification of the 16S gene were designed based on the con-served regions in the 518F, 800R, 27F, and 1492R genome sequences [18].“
[19] instead of [18]
2. „Sequencing of extracted DNA was carried out by the Infection Disease Unit, King Fahad Medical Center for Research, King Abdulaziz University 8/2/2021, the sequenced data were analyzed using BLAST-NCBI“
What exact type of Illumina sequencer did you use for the sequencing? How long (how many bp) were the reads obtained after sequencing?
3. „The phylogenetic tree diagram of the 7 isolated MDR bacteria constructed using the maximum likelihood (ML) algorithm in MEGA7 software (megasoftware.net (accessed on 2 September 2021)).“
7 MDR bacteria? 75 MDR bacterial isolates you mentioned in the chapter „2.3. Isolation of Multidrug-Resistant (MDR) Bacteria“?
4. I would recommend dividing the chapter „16S rRNA Gene Sequence and Phylogenetic Analysis“ into 2
the first one: 16S rRNA Gene Sequence
and the second one: Phylogenetic Analysis – this one should be after the chapter „Sequencing and Analysis Workflow“
I also recommend to rearrange the chapters according to the work flow of this whole study:
1. DNA extraction
2. sequecing
3. bioinfo analysis as MDR seq.assembly and phylogenetic trees
So e.g. chapter „2.5. De Novo Assembly of MDR Bacterial Sequences“ should be after the chapter „Sequencing and Analysis Workflow“
2.6. Sequencing and Analysis Workflow
„Sequencing: For library construction, DNA was extracted from the isolated bacteria according to an appropriate protocol, depending on the 7 isolated bacteria, to obtain the best results.“
According to what protocol? It would be appropriate to mention exactly what protocols you followed and whether you made any modifications.
2. You are describing here the general workflow of Ilumina seq.platform – no need. How did you specifically do your analysis? What method did you use for DNA fragmentation and how long were the fragments obtained? What exact type of Ilumina sequencer did you use for the sequencing? How long (how many bp) were the reads obtained after sequencing? Etc.
Results
Rewrite the whole section, use the tables, divide the section into chapters such as 16S analysis, detected determinants, phylogenetic tree analysis, ..
What was the point of creating the phylogenetic tree for different bacterial species based on individual contigs? Should not you use the whole genomes for this kind of analysis? What were you trying to get at?
Table containing info about which resistance genes at which bacterial species you detected. What type of OXA, CTX-M exactly? Etc.
Author Response
Introduction: Rewrite the whole section – it is not necessary to describe the general mechanism of action of beta-lactam antibiotics and general resistance mechanisms + citations are shifted:
- „Regarding the Gram-positive MDR bacteria, a study conducted on the incidence of Gram[1]positive MDR bacteria at King Abdulaziz University Hospital (KAUH), Jeddah, found that the most prevalent isolates were S. aureus (MRSA), at 20% [15]“
Check the citations and reference list, it appears that citations are somehow shifted. It occurs to me that instead of [15]“there should be [17]“ etc.
Thank you for the valuable information; we have checked and corrected the reference and citation.
Materials and methods:
2.2 Clinical specimens How many isolates did you have in total – in abstract, you mentioned 108 – it needs to be mentioned also here. How many have you sequenced? 75 or 7? It is not clear from methods. How many bacteria were analysed for phylogenetic tree and for the resistance determinants analysis? How did you determine the susceptibility of isolates? You mentioned Phoenix BD system in abstract but there is no information here, in methods.
In light of these comments, we have specified the number of isolates in the Materials and Methods section and described them as well as the MDR isolates. We also added the susceptibility determination to this section.
2.3. Isolation of Multidrug-Resistant (MDR) Bacteria
Should be mentioned in results. Done.
2.4. 16S rRNA Gene Sequence and Phylogenetic Analysis:
- Correct the citations, e.g.:
„Bacterial DNA was extracted using the Qiagen DNA extract kit as per manufactur-er’s instructions [16].“
[18] instead of [16] Thank you; the citation has been corrected.
„The isolated DNA samples were stored at −20 °C according to the manufacturer’s protocol, as outlined in the DNeasy Blood and Tissue Handbook [17].“
[18] instead of [17]...or it is not necessary to mention this handbook again – generally is extracted DNA kept in -20. Done.
„The primers for the amplification of the 16S gene were designed based on the con-served regions in the 518F, 800R, 27F, and 1492R genome sequences [18].“
[19] instead of [18] Thank you; the citation has been corrected.
- „Sequencing of extracted DNA was carried out by the Infection Disease Unit, King Fahad Medical Center for Research, King Abdulaziz University 8/2/2021, the sequenced data were analyzed using BLAST-NCBI“
What exact type of Illumina sequencer did you use for the sequencing? How long (how many bp) were the reads obtained after sequencing?
Thank you for your pointing this out.
- „The phylogenetic tree diagram of the 7 isolated MDR bacteria constructed using the maximum likelihood (ML) algorithm in MEGA7 software (megasoftware.net (accessed on 2 September 2021)).“
7 MDR bacteria? 75 MDR bacterial isolates you mentioned in the chapter „2.3. Isolation of Multidrug-Resistant (MDR) Bacteria“?
Thank you for your the question, of the 108 isolates, 75 were MDR, among which we selected 7.
- I would recommend dividing the chapter „16S rRNA Gene Sequence and Phylogenetic Analysis“ into 2
the first one: 16S rRNA Gene Sequence and
the second one: Phylogenetic Analysis – this one should be after the chapter „Sequencing and Analysis Workflow“
Thank you for your this recommendation. The phylogenetic tree is part of the sequencing and method of analyzing just one line.
I also recommend to rearrange the chapters according to the work flow of this whole study:
- DNA extraction
- sequecing
- bioinfo analysis as MDR seq.assembly and phylogenetic trees
So e.g. chapter „2.5. De Novo Assembly of MDR Bacterial Sequences“ should be after the chapter „Sequencing and Analysis Workflow“ 2.6. Sequencing and Analysis Workflow
Thank you for your these valuable recommendations, which we have followed.
- „Sequencing: For library construction, DNA was extracted from the isolated bacteria according to an appropriate protocol, depending on the 7 isolated bacteria, to obtain the best results.“
According to what protocol? It would be appropriate to mention exactly what protocols you followed and whether you made any modifications.
In light of this recommendation, we describe the protocol in the revised version of section 2.4 Extraction and Sequencing of the 16S rRNA Gene and Phylogenetic Analysis. The sentence revised and highlighted.
- You are describing here the general workflow of Ilumina seq.platform – no need.
How did you specifically do your analysis? What method did you use for DNA fragmentation and how long were the fragments obtained? What exact type of Ilumina sequencer did you use for the sequencing? How long (how many bp) were the reads obtained after sequencing? Etc.
Thank you for this valuable recommendation. Changes in the methods have done.
Results:
Rewrite the whole section, use the tables, divide the section into chapters such as 16S analysis, detected determinants, phylogenetic tree analysis, ..
Thank you for your these valuable recommendations, which we have followed.
What was the point of creating the phylogenetic tree for different bacterial species based on individual contigs? Should not you use the whole genomes for this kind of analysis? What were you trying to get at?
In light of this recommendation, we have constructed a phylogenetic tree for our isolates with the comparison of the most similar neighborhood according to the contig, we added the contig number in each strain (we chose the seven most similar strains).
Table containing info about which resistance genes at which bacterial species you detected. What type of oxa, ctxm exactly? Etc.
Thank you for this valuable recommendation, There are now seven supplementary tables that describe the resistance genes for each bacteria (S1-S7).
We found the oxa gene in E. coli and S. aureus, roxA in G- ([50S ribosomal protein L16]-arginine 3-hydroxylase), and qoxA in G+ (cytochrome aa3 quinol oxidase subunit II). We found the ctxm gene in K. pneumonia.
Round 2
Reviewer 1 Report
Albeit the authors do make an effort to improve their manuscript, they fail to address the key flaws indicated in the first report. Most edits are only superficial and din not drastically improved the quality of the manuscripts.
1. The methodology section is still vague and unprofessionally written. Key pieces of information essential for validation of the results are still missing and are still investing a lot of space in excerpts from well know manufacturing protocols.
2. Deposition of the sequencing data (both reads and contigs) to an appropriate database is mandatory these days for any genomic study. The authors failed to deposit the data, or if they did, they fail to clearly indicate the SRA entries and the associated contigs.
3. Figures are still mislabeled. The general Figure 1, albeit not bringing any novelty in the study is still present. Again, this figure should be used as a general guide to building a proper methods section, with technical details on the bioinformatics analysis. As it is, it has no point or significance as it does not allow the reader to replicate the study. Despite the indicated resolution being consistent with the journal’s instructions , the figures are still unreadable. I for example can not read most of the text in the figures, including scales in graphs. That is why I am unable to evaluate the results.
4. Discussions are still very general and still very seldom make reference to the data presented here. The authors decided to add more references instead, and not to properly and extensively discuss their data.
5. Comment 5 was extensively addressed.
Author Response
1. The methodology section is still vague and unprofessionally written. Key pieces of information essential for validation of the results are still missing and are still investing a lot of space in excerpts from well know manufacturing protocols.
We submitted our response in revision 1, the sequences and the analysis were done by Macrogen and we contact the company to answer reviewers questions, if there is particular question you need it's answer, please just ask it in particular. Your comment itself is vague.
3. Deposition of the sequencing data (both reads and contigs) to an appropriate database is mandatory these days for any genomic study. The authors failed to deposit the data, or if they did, they fail to clearly indicate the SRA entries and the associated contigs.
Thank you for your comment as we previously responded, we can remove it, because our article aimed to find resistant genes and new isolation of MDR bacteria, we use the methodology provided by Macrogen and respective references, and it’s not our aim of this article to find novel method.
4. Figures are still mislabeled. The general Figure 1, albeit not bringing any novelty in the study is still present. Again, this figure should be used as a general guide to building a proper methods section, with technical details on the bioinformatics analysis. As it is, it has no point or significance as it does not allow the reader to replicate the study. Despite the indicated resolution being consistent with the journal’s instructions, the figures are still unreadable. I for example can not read most of the text in the figures, including scales in graphs. That is why I am unable to evaluate the results.
We understand your comments, all the figures were added as the followed in journal’s instructions, we submitted the manuscript and our response to the reviewers, to a respected editing website for 3rd time editing and evaluating.
5. Discussions are still very general and still very seldom make reference to the data presented here. The authors decided to add more references instead, and not to properly and extensively discuss their data.
The discussion covered all the results provided in the results section, as one of the reviewers recommended discuss the other genes were found in our isolates, we have added them. If you have particular part to mention in the discussion section, please name it.
I attached pdf report of E. coli result from Macrogen.

Reviewer 3 Report
Dear authors,
Based on my evaluation, I did not observe significant improvement in the new version of the manuscript. Therefore, I advise against publishing it in Microorganisms. However, I do have some suggestions to enhance the manuscript.
Please, use decimal notation instead of whole numbers for percentages. For example, write "32.0%" instead of "32%" when the percentage is a whole number. Don't forget to check and apply this formatting throughout the entire text.
You should make a supplementary table with all the antimicrobial agents used and their susceptibility percentage.
Abstract
Replace ‘75 of which (69.44%) were MDR’ for ‘75 (69.44%) of which were MDR’
Material and Methods
In the Results section you mention phenotype analysis and biochemical testing used to identify the bacteria, but in the Material and Methods you mention the phenotype identification without information regarding the biochemical test. Please, describe the biochemical test in the Material and Methods. Also, inform the antimicrobial agent’s names that were tested against the isolates, if the list is long, inform the how many antimicrobial agents of each class were used. For example: Beta lactams (n=6), fluoroquinolones (n=2), Aminoglycosides (n=3)
2.2. Clinical Specimens
In the first report I asked you to add the number of MDR isolates in the Material and Methods, but reading the manuscript again I see that it is better to remove the information regarding de 75 MDR isolates. Actually, the number of total isolates and MDR should be in the Result section.
This sub-section should be as following: ‘Bacterial strains were isolated from different types of medical specimens (stool, sputum, blood, urine, high vaginal swabs (HVS), and swabs from wounds, ears, eyes, noses, etc.) and from several hospitals wards, e.g., the neonatal intensive care unit (NICU), the intensive care unit (ICU), the emergency room (ER), and the outpatient departments (OPD). The isolates were collected from three hospitals (Ministry of Health)between December 2018 and November 2019. All of the isolates from each hospital laboratory were grown on blood agar and then transferred to the microbiology laboratory at Jeddah University for further studies.’
Move and rearrange the following information to the Result section: ‘Among a total of 108 isolates analyzed for this study, 75 were MDR bacteria’ and ‘In terms of the age of the patients involved, the number of bacteria isolates were 41 (37.97%) from those aged 0-10 years, 10 (9.26%) from those aged 11-20 years, 20 (18.52%) from those aged 21-30 years, 19 (17.59%) from those aged 31-40 years, and 18 (16.67%) from those over the age of 41. In terms of gender, 32 (29.63%) of the isolates were from male patients and 76 (70.37%) were from female patients.’
2.3. Isolation of Multidrug-Resistant (MDR) Bacteria:
The following sentence should be in the Result section:
‘Of the 108 isolates, 75 (69.44%) were identified as MDR bacteria. These isolates were found to be predominately represented by the following strains: S. aureus (MRSA), 17 (22.67%); multidrug-resistant S. epidermidis, 1 (1.33%); multidrug-resistant S. pneumoniae, 2 (2.67%); ESBL-producing Gram-negative Enterobacteriaceae such as ESBL E. coli, 24 (32%); ESBL K. pneumoniae, 19 (25.33%); E. cloacae, 2 (2.67%); and multidrug-resistant P. aeruginosa, 9 (12%). Two isolates each of E. coli and K. pneumoniae and three isolates of S. aureus were selected for further study.’
Replace ‘Two isolates each of E. coli and K. pneumoniae and three isolates of S. aureus were selected for further study’ for ‘We selected two isolates each of E. coli and K. pneumoniae, as well as three isolates of S. aureus to perform the remaining tests.’
4. Results
Since you only tested seven isolates to identify resistance genes, I still think you should make a table showing the resistance genes found in these isolates.
4.1 Identification of MDR Isolates
Here the title should be ‘Identification of the bacterial strains’, because the first information you added in this sub-section is about the species identified.
Then, you should add a sub-section ‘Antimicrobial susceptibility testing and Identification of MDR Isolates’. Here you write the how many of the 108 isolates were susceptible to all antimicrobial agents used, the most common resistances, for example: tetracycline (93.7%), nalidixic acid (81.3%), and ampicillin (50.0%). If it is more convenient, write the most common resistance of the antimicrobial agent classes as I said previously.
After it you add ‘Of the 108 isolates, 75 (69.44%) were identified as MDR bacteria. These isolates were found to be predominately represented by the following strains: S. aureus (MRSA), 17 (22.67%); multidrug-resistant S. epidermidis, 1 (1.33%); multidrug-resistant S. pneumoniae, 2 (2.67%); ESBL-producing Gram-negative Enterobacteriaceae such as ESBL E. coli, 24 (32%); ESBL K. pneumoniae, 19 (25.33%); E. cloacae, 2 (2.67%); and multidrug-resistant P. aeruginosa, 9 (12%). We selected two isolates each of E. coli and K. pneumoniae, as well as three isolates of S. aureus to perform the remaining tests’ - Please, check the number of MDR isolates, by my calculation there are 74 MDR isolates described here
Keep in mind that MDR isolates must be resistant to drugs from three or more classes of antimicrobial agents and not simply be resistant to three drugs, as you stated in the following sentence ‘AST results identified one isolate of P. aeruginosa, three isolates of E. coli, four of K. Pneumoniae, and three of S. aureus as being resistant to more than three antibiotics. Seven of these isolates were selected for the detection of MDR genes.’
Why you chose only seven isolates for detection of MDR genes if you had 75 MDR isolates? Explain in the text the reason.
5. Discussion
The discussion is still too general. When comparing your results with another study, it is important to provide relevant information about the study being compared, such as the country where the study was performed, the source of the isolates (bloodstream infection, urinary tract infection, lung infection), and any other key findings, to give context to your own findings.
You should discuss the results of the susceptibility test (since you are going to add this information in the Material and Methods and Results section).
Author Response
Please, use decimal notation instead of whole numbers for percentages. For example, write "32.0%" instead of "32%" when the percentage is a whole number. Don't forget to check and apply this formatting throughout the entire text.
Done.
You should make a supplementary table with all the antimicrobial agents used and their susceptibility percentage.
There is a supplementary table for antibiotics and the resistant of isolated bacteria.
Abstract
Replace ‘75 of which (69.44%) were MDR’ for ‘75 (69.44%) of which were MDR’
Thank you, we have made these changes.
Materials and Methods
In the Results section you mention phenotype analysis and biochemical testing used to identify the bacteria, but in the Material and Methods you mention the phenotype identification without information regarding the biochemical test. Please, describe the biochemical test in the Material and Methods. Also, inform the antimicrobial agent’s names that were tested against the isolates, if the list is long, inform the how many antimicrobial agents of each class were used. For example: Beta lactams (n=6), fluoroquinolones (n=2), Aminoglycosides (n=3)
The identification of the isolates was performed by phoenix BD, and this identification includes biochemical and antimicrobial activity. The word “biochemical” was deleted from the results section.
We checked antimicrobial activity against 108 universal antibiotics (performed by Macrogen), these were added to the supplementary table.
2.2. Clinical Specimens
In the first report I asked you to add the number of MDR isolates in the Material and Methods, but reading the manuscript again I see that it is better to remove the information regarding de 75 MDR isolates. Actually, the number of total isolates and MDR should be in the Result section.
Thank you, we have made these changes.
This sub-section should be as following: ‘Bacterial strains were isolated from different types of medical specimens (stool, sputum, blood, urine, high vaginal swabs (HVS), and swabs from wounds, ears, eyes, noses, etc.) and from several hospitals wards, e.g., the neonatal intensive care unit (NICU), the intensive care unit (ICU), the emergency room (ER), and the outpatient departments (OPD). The isolates were collected from three hospitals (Ministry of Health) between December 2018 and November 2019. All of the isolates from each hospital laboratory were grown on blood agar and then transferred to the microbiology laboratory at Jeddah University for further studies.’
Thank you, we have made these changes.
Move and rearrange the following information to the Result section: ‘Among a total of 108 isolates analyzed for this study, 75 were MDR bacteria’ and ‘In terms of the age of the patients involved, the number of bacteria isolates were 41 (37.97%) from those aged 0-10 years, 10 (9.26%) from those aged 11-20 years, 20 (18.52%) from those aged 21-30 years, 19 (17.59%) from those aged 31-40 years, and 18 (16.67%) from those over the age of 41. In terms of gender, 32 (29.63%) of the isolates were from male patients and 76 (70.37%) were from female patients.’
Thank you, we have made these changes.
2.3. Isolation of Multidrug-Resistant (MDR) Bacteria:
The following sentence should be in the Result section:
‘Of the 108 isolates, 75 (69.44%) were identified as MDR bacteria. These isolates were found to be predominately represented by the following strains: S. aureus (MRSA), 17 (22.67%); multidrug-resistant S. epidermidis, 1 (1.33%); multidrug-resistant S. pneumoniae, 2 (2.67%); ESBL-producing Gram-negative Enterobacteriaceae such as ESBL E. coli, 24 (32%); ESBL K. pneumoniae, 19 (25.33%); E. cloacae, 2 (2.67%); and multidrug-resistant P. aeruginosa, 9 (12%). Two isolates each of E. coli and K. pneumoniae and three isolates of S. aureus were selected for further study.’
Thank you, we have made these changes.
Replace ‘Two isolates each of E. coli and K. pneumoniae and three isolates of S. aureus were selected for further study’ for ‘We selected two isolates each of E. coli and K. pneumoniae, as well as three isolates of S. aureus to perform the remaining tests.’
Thank you, we have made these changes.
- Results
Since you only tested seven isolates to identify resistance genes, I still think you should make a table showing the resistance genes found in these isolates.
The table is added as a supplementary table.
4.1 Identification of MDR Isolates
Here the title should be ‘Identification of the bacterial strains’, because the first information you added in this sub-section is about the species identified.
Thank you, we have made the changes.
Then, you should add a sub-section ‘Antimicrobial susceptibility testing and Identification of MDR Isolates’. Here you write the how many of the 108 isolates were susceptible to all antimicrobial agents used, the most common resistances, for example: tetracycline (93.7%), nalidixic acid (81.3%), and ampicillin (50.0%). If it is more convenient, write the most common resistance of the antimicrobial agent classes as I said previously.
The aim of the study is to compare de novo and pan-genomic analysis. We have results of the antibiotic susceptibility test for antibiotics used by the MOH in Saudi Arabia and the global one, but we did not plan to add them because we want to focus on the resistant genes using two methods. The antibiotics results can be published in another article as a result of antibiogram.
After it you add ‘Of the 108 isolates, 75 (69.44%) were identified as MDR bacteria. These isolates were found to be predominately represented by the following strains: S. aureus (MRSA), 17 (22.67%); multidrug-resistant S. epidermidis, 1 (1.33%); multidrug-resistant S. pneumoniae, 2 (2.67%); ESBL-producing Gram-negative Enterobacteriaceae such as ESBL E. coli, 24 (32%); ESBL K. pneumoniae, 19 (25.33%); E. cloacae, 2 (2.67%); and multidrug-resistant P. aeruginosa, 9 (12%). We selected two isolates each of E. coli and K. pneumoniae, as well as three isolates of S. aureus to perform the remaining tests’ - Please, check the number of MDR isolates, by my calculation there are 74 MDR isolates described here
The total number of MDR bacteria was revised “K. aerogenes 1 (1.33%)” was missed.
Keep in mind that MDR isolates must be resistant to drugs from three or more classes of antimicrobial agents and not simply be resistant to three drugs, as you stated in the following sentence ‘AST results identified one isolate of P. aeruginosa, three isolates of E. coli, four of K. Pneumoniae, and three of S. aureus as being resistant to more than three antibiotics. Seven of these isolates were selected for the detection of MDR genes.’
Yes, the identification of MDR is correct. They are resistant to more than three classes of antibiotics.
The sentence was corrected, and classes were unwritten by mistake.
Why you chose only seven isolates for detection of MDR genes if you had 75 MDR isolates? Explain in the text the reason.
Thank you, we have made the changes.
- Discussion
The discussion is still too general. When comparing your results with another study, it is important to provide relevant information about the study being compared, such as the country where the study was performed, the source of the isolates (bloodstream infection, urinary tract infection, lung infection), and any other key findings, to give context to your own findings.
The aim of the study is the comparison between de novo and pan-genomic,
You should discuss the results of the susceptibility test (since you are going to add this information in the Material and Methods and Results section).
We did not add the susceptibility test in materials and methods or results.
Reviewer 4 Report
The manuscript was significantly improved based on my comments. However, I believe that an article of this focus is much more appropriate for a journal focused on local data. I am afraid that the data presented will not be of interest to readers from other geographical regions.
So, unfortunately, I cannot recommend the article for acceptance.
Author Response
The manuscript was significantly improved based on my comments. However, I believe that an article of this focus is much more appropriate for a journal focused on local data. I am afraid that the data presented will not be of interest to readers from other geographical regions.
So, unfortunately, I cannot recommend the article for acceptance.
Thank you so much for your comments, but I was wondering about your recommendation that the data presented will not be of interest to readers from other geographical regions?! The aim of the study is to find whether there are new strains of MDR bacteria and to find the resistance genes in the isolated bacteria. This is not reflective of the geographical regions, but on global and human health. We saw an example of this during the COVID-19 pandemic. An article is needed on this subject or the journal will not provide a special issue on this topic! Below, you can find 6 published articles, two of which were published in 2023:
- Szekeres, E., Baricz, A., Chiriac, C. M., Farkas, A., Opris, O., Soran, M. L., ... & Coman, C. (2017). Abundance of antibiotics, antibiotic resistance genes and bacterial community composition in wastewater effluents from different Romanian hospitals. Environmental Pollution, 225, 304-315.
- Ghenea, A. E., Zlatian, O. M., Cristea, O. M., Ungureanu, A., Mititelu, R. R., Balasoiu, A. T., ... & Balasoiu, M. (2022). TEM, CTX-M, SHV Genes in ESBL-Producing Escherichia coli and Klebsiella pneumoniae Isolated from Clinical Samples in a County Clinical Emergency Hospital Romania-Predominance of CTX-M-15. Antibiotics, 11(4), 503.
- Krüger, H., Ji, X., Hanke, D., Schink, A. K., Fiedler, S., Kaspar, H., ... & Feßler, A. T. (2022). Novel macrolide-lincosamide-streptogramin B resistance gene erm (54) in MRSA ST398 from Germany. Journal of Antimicrobial Chemotherapy, 77(8), 2296-2298.
- Szymanek-Majchrzak, K., & MÅ‚ynarczyk, G. (2022). Genomic Insights of First erm B-Positive ST338-SCC mec VT/CC59 Taiwan Clone of Community-Associated Methicillin-Resistant Staphylococcus aureu s in Poland. International Journal of Molecular Sciences, 23(15), 8755.
- Bedenić, B., Bratić, V., Mihaljević, S., Lukić, A., Vidović, K., Reiner, K., ... & Grisold, A. (2023). Multidrug-Resistant Bacteria in a COVID-19 Hospital in Zagreb. Pathogens, 12(1), 117.
- Álvarez-Lerma, F., Catalán-González, M., Álvarez, J., Sánchez-García, M., Palomar-Martínez, M., Fernández-Moreno, I., ... & Martínez-Alonso, M. (2023). Impact of the “Zero Resistance” program on acquisition of multidrug-resistant bacteria in patients admitted to Intensive Care Units in Spain. A prospective, intervention, multimodal, multicenter study. Medicina Intensiva (English Edition).
Also, you can check the attached file of the related papers published in MDPI journals, which added in my submission page. You can see 3 of the 5 articles are local isolation bacteria?!
